# Somatic mutations in clonally expanded cytotoxic T lymphocytes in patients with newly diagnosed rheumatoid arthritis

P. Savola[1,2,*], T. Kelkka[1,2,*], H.L. Rajala[1], A. Kuuliala[3], K. Kuuliala[3], S. Eldfors[4], P. Ellonen[4], S. Lagström[4], M. Lepistö[4], T. Hannunen[4], E.I. Andersson[1], R.K. Khajuria[1], T. Jaatinen[5], R. Koivuniemi[6], H. Repo[3], J. Saarela[4], K. Porkka[1,**], M. Leirisalo-Repo[6,**] & S. Mustjoki[1,2,**]

Somatic mutations contribute to tumorigenesis. Although these mutations occur in all proliferating cells, their accumulation under non-malignant conditions, such as in autoimmune disorders, has not been investigated. Here, we show that patients with newly diagnosed rheumatoid arthritis have expanded CD8+ T-cell clones; in 20% (5/25) of patients CD8+ T cells, but not CD4+ T cells, harbour somatic mutations. In healthy controls ($n = 20$), only one mutation is identified in the CD8+ T-cell pool. Mutations exist exclusively in the expanded CD8+ effector-memory subset, persist during follow-up, and are predicted to change protein functions. Some of the mutated genes (*SLAMF6, IRF1*) have previously been associated with autoimmunity. RNA sequencing of mutation-harbouring cells shows signatures corresponding to cell proliferation. Our data provide evidence of accumulation of somatic mutations in expanded CD8+ T cells, which may have pathogenic significance for RA and other autoimmune diseases.

[1] Hematology Research Unit Helsinki, University of Helsinki and Department of Hematology, Helsinki University Hospital Comprehensive Cancer Centre, Haartmaninkatu 8, FIN-00290 Helsinki, Finland. [2] Department of Clinical Chemistry and Hematology, University of Helsinki, Haartmaninkatu 8, FIN-00290 Helsinki, Finland. [3] Bacteriology and Immunology, Medicum, University of Helsinki, Haartmaninkatu 3, FIN-00290 Helsinki, Finland. [4] Institute for Molecular Medicine Finland (FIMM), University of Helsinki, Tukholmankatu 8, FIN-00290 Helsinki, Finland. [5] Histocompatibility Testing Laboratory, Finnish Red Cross Blood Service, Kivihaantie 7, FIN-00310 Helsinki, Finland. [6] Department of Rheumatology, University of Helsinki and Helsinki University Hospital, Haartmaninkatu 4, FIN-00290 Helsinki, Finland. * These authors contributed equally to this work. ** These authors jointly supervised this work. Correspondence and requests for materials should be addressed to S.M. (email: satu.mustjoki@helsinki.fi).

T cells undergo clonal expansion after they recognize cognate antigens, which are presented by professional antigen-presenting cells. Because the mutation rate in lymphoid precursors has been estimated to be $\sim 10^{-6}$ per cell division, errors in DNA replication are likely to accumulate during cell divisions[1,2]. Somatic mutations have a critical function in cancer pathogenesis, and mutations derived from haematopoietic stem cells and progenitor cells have also been detected in peripheral-blood cells. This phenomenon is called clonal haematopoiesis[3–7]. The occurrence of somatic mutations in mature, clonally expanded, non-malignant T-cell populations has not been fully investigated, nor have their effects on the disease process and clonal properties been determined.

Rheumatoid arthritis (RA) is a common systemic inflammatory disease of autoimmune origin[8], characterized by chronic inflammation[9], and T-cell-mediated autoantibody production occurring years before the clinical diagnosis[10–12]. RA often co-manifests in patients with large granular lymphocyte (LGL) leukaemia, a rare lymphoproliferative disorder in which we recently discovered that expanded CD8+ T-cell clones harbour somatic STAT3 mutations[13]. Interestingly, LGL leukaemia patients with multiple STAT3 mutations have a higher incidence of RA (43%) than patients without STAT3 mutations (6%)[14], thus raising the possibility that patients with STAT3-mutated LGL leukaemia and clonal-lymphocyte proliferation are at a greater risk of developing a non-malignant autoimmune disease and that CD8+ T-cell clones carrying somatic mutations may participate in the autoimmune process. Alternatively, autoimmune diseases may facilitate the acquisition of somatic mutations in lymphocytes.

In this study, we investigate whether T cells in people with a non-malignant autoimmune disorder without known lympho-proliferation harbour somatic mutations and analyse blood samples from individuals with newly diagnosed, untreated RA. With a flow-cytometry-based T-cell clonality screen we identify potential clonal expansions in the CD8+ T cells and confirm the findings using deep T-cell receptor β chain (TCRB) sequencing. We discover somatic mutations in purified CD8+ T cells with a custom deep-sequencing panel including 986 immune-related genes and with exome and deep amplicon sequencing. These results provide a link between cancerous processes and autoimmunity.

## Results

**The clonality of CD8+ T cells increases with age.** We collected blood samples from 82 untreated newly diagnosed ACR/EULAR 2010 criteria-fulfilling RA patients and 20 healthy controls after obtaining written informed consent (Fig. 1a). T-cell clonality was initially screened with a flow-cytometry-based assay using a panel of TCR Vβ-specific antibodies. This method allows for rapid and robust analysis of TCRB V-gene usage and facilitates flow-cytometry-based cell sorting of the populations of interest. To assess the clonality in more detail (a T-cell clone is defined by a unique TCR β complementarity-determining region 3 [CDR3] sequence), the samples were subjected to a next-generation sequencing (NGS)-based TCR β (TCRB CDR3) deep-sequencing assay.

Flow cytometry screening indicated increased clonality in CD8+ T cells compared with CD4+ T cells (Supplementary Fig. 1). Next, sorted CD8+ fractions were subjected to NGS TCRB sequencing. The CD8+ V-gene usage in flow cytometry and TCRB-sequencing correlated well (Fig. 1b; $P < 0.0001$, Spearman correlation coefficient $r = 0.61$), thus justifying the use of flow cytometry as a screening and sorting method.

Deep TCRB sequencing showed that in RA patients, the clonality of CD8+ T cells (measured as the productive clonality index) did not significantly differ from that in age-matched healthy controls (Fig. 1c). Interestingly, in patients, but not in healthy controls, the clonality increased with age (Fig. 1d; $P < 0.0001$, Spearman correlation coefficient $r = 0.49$; Supplementary Fig. 2).

A proportion of RA patients harboured prominent CD8+ T-cell clones. In 8/65 patients the largest clone composed over 20% of the total CD8+ cells, and in one individual patient (Pt 3) the largest clone composed 51% of the total CD8+ lymphocyte pool (Table 1, Fig. 1e). In the healthy controls, we identified only two individual CD8+ clones that exceeded 20% of the total CD8+ cells (Fig. 1e), whereas the largest clone in healthy controls accounted for 26% (Table 1). In addition to clone size, the V-gene usage distribution of clones composing over 1% of the CD8+ cells was compared between the healthy controls and RA patients. Only TCRBV07 was slightly overrepresented in the healthy controls, and no differences were observed in other V genes (Fig. 1f,g). RA patients' TCRB amino-acid sequences were not shared among CD8+ clones larger than 1%, with some exceptions (Supplementary Table 1).

The CD8+ T-cell clonality was correlated with clinical parameters (Table 1 and Supplementary Table 2) to discover possible associations (Supplementary Figs 3 and 4). The CD4/CD8-ratio (Spearman correlation coefficient $r = -0.52$), the number of swollen joints (Spearman $r = -0.41$) and haemoglobin level (Spearman $r = -0.26$) were each statistically significantly correlated with the CD8+ T-cell clonality.

From two patients (6 and 23, Table 1), paired synovial fluid and peripheral-blood samples were available. Several identical CD8+ TCRB sequences were detected in the peripheral blood and synovial fluid (Supplementary Fig. 5).

HLA-A, -B, -C and –DRB1 haplotyping was performed on 65 patients and 20 controls (Supplementary Table 3). The HLA-DRB1 allele frequencies demonstrated that our RA patients shared the genetic characteristics typical of RA: the shared epitope[15] (RA risk allele) frequencies were higher in RA patients than in healthy controls (Supplementary Fig 6; Fisher's test $P = 0.03$, OR 2.32). HLA-I allele frequencies did not differ between the patients and the controls (Supplementary Fig. 7). Further, when the patients were divided into two groups on the basis of the median clonality index (high versus low clonality), no statistically significant differences were observed in shared epitope or HLA-I allele frequencies.

**Somatic mutations are identified in CD8+ cells.** After analysing T-cell clonality, we assessed whether somatic mutations existed in purified CD8+ and/or CD4+ cells (Fig. 1a). We designed a custom-sequencing panel of 986 immune-related genes (Fig. 1a; a list of sequenced genes is presented in Supplementary Table 4) to achieve greater sequencing depth ($500-1,000 \times$) than exome sequencing ($50-100 \times$). This methodology enabled us to detect mutations with lower variant allele frequencies (VAFs).

Immunogene panel sequencing was performed on sorted CD8+ and CD4+ cells from 25 patients (median age 61, range 19–75) and 20 healthy controls (median age 55, range 21–70). For one patient (Pt 23, Table 1), the sequencing assay was performed using CD4+ and CD8+ cells isolated from synovial fluid. A robust and verified variant-calling bioinformatics pipeline was used for mutation calling[13]. Each patient's own CD8+ cells were compared with his/her own CD4+ cells, and a list of variants occurring only in CD8+ (and not in CD4+) was obtained. Thus, variants identified in both cell fractions were excluded from further analyses. This approach allowed for exclusion of germline variants and variants occurring in haematopoietic progenitor

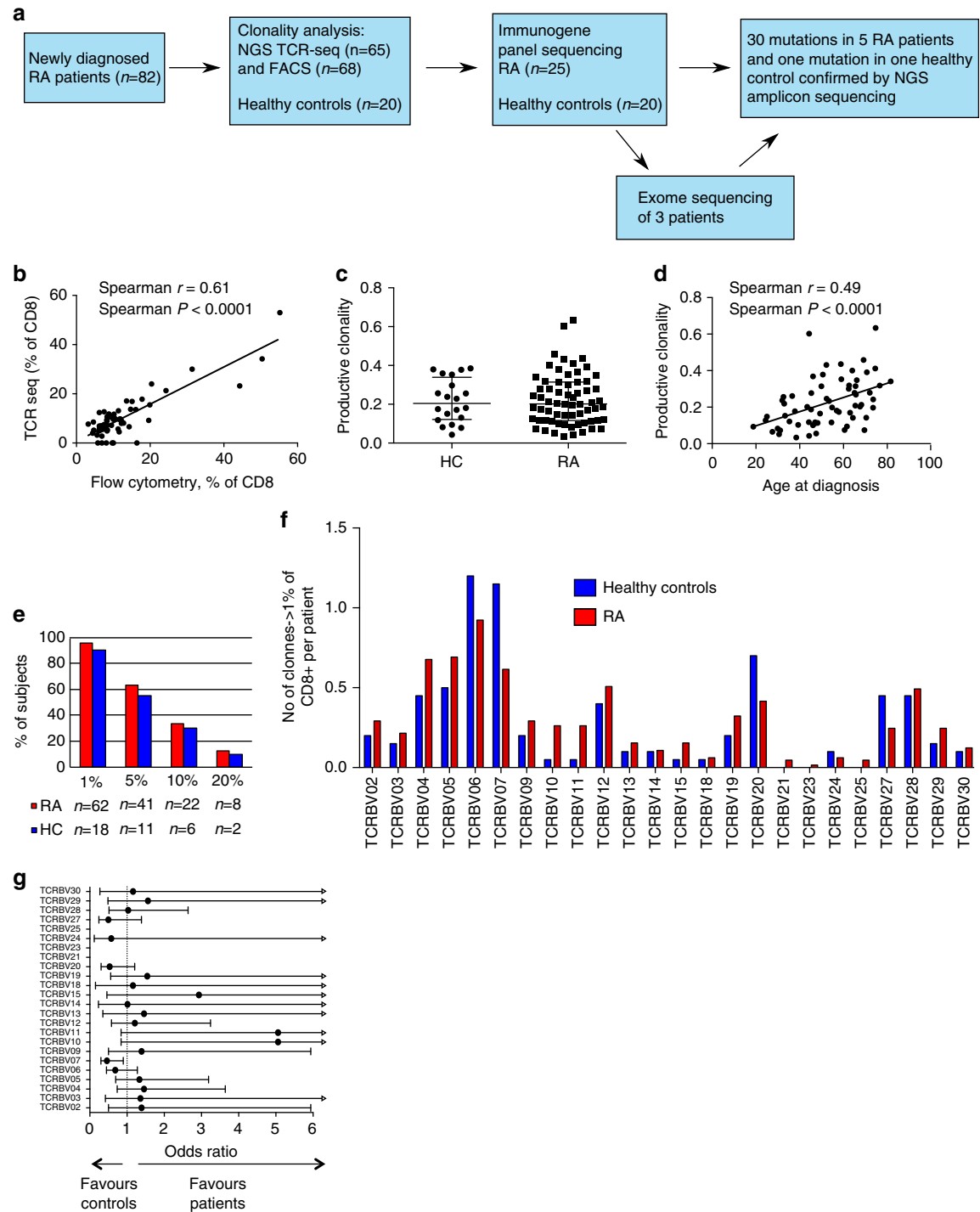

**Figure 1 | Clonal expansions of CD8 + T cells occur in RA patients and clonality increases with age.** (**a**) The study workflow, presented as a flow chart. (**b**) The V-gene usage of the largest flow-antibody-stained population correlated well with the corresponding V-gene usage in TCRB sequencing. The comparison was performed using the 61 patients for whom both flow cytometry and TCRB-sequencing data were available. Correlation was tested with Spearman correlation. (**c**) CD8 + clonality index calculated on the basis of TCRB sequencing was compared between patients ($n = 65$) and healthy controls (HC, $n = 20$) with the Mann–Whitney test, but there was no statistically significant difference. The horizontal line represents the group median, and error bars interquartile range (IQR). (**d**) The CD8 + TCRB-sequencing-based clonality index increases with age in RA patients (tested with Spearman correlation, $n = 65$). (**e**) The percentage of patients and controls harbouring at least one CD8 + clone larger than 1%, 5%, 10%, and 20% of CD8 + is shown. (**f**) The V-gene usage in CD8 + clones comprising over 1% of all CD8 + in patients and controls is shown. The data were obtained via TCRB sequencing, and are presented as the number of clones using the denoted TCRBV gene adjusted for the number of individuals (RA = 65, HC = 20). (**g**) Odds ratios for CD8 + clones comprising over 1% of CD8 + for each TCRBV gene. The error bars represent 95% confidence intervals (CI), the arrow showing that the upper limit does not fit the scale. There were no differences between patients and controls (tested with Fisher's test), except for TRBV07, which may be slightly underrepresented in patients compared with controls (Fisher's test, $P = 0.00664$, not adjusted for multiple comparisons, error bars 95% CI).

**Table 1 | Clinical characteristics of patients and healthy controls sequenced with immunogene panel sequencing.**

| Patient ID | Sex | Age at diagnosis | Seropositive | DAS28 | Shared epitope | Clonality | Largest CD8+ clone (%) |
|---|---|---|---|---|---|---|---|
| 1 | M | 75 | Yes | 3.5 | Yes/no | 0.63 | 33.9 |
| 2 | F | 72 | Yes | 3.5 | Yes/yes | 0.28 | 7.8 |
| 3 | M | 44 | Yes | NA | Yes/no | 0.60 | 51.0 |
| 4 | F | 74 | No | NA | No/no | 0.24 | 13.2 |
| 5 | F | 59 | Yes | NA | No/no | 0.44 | 29.0 |
| 6 | M | 75 | Yes | 4.0 | Yes/yes | 0.41 | 11.1 |
| 7 | M | 66 | Yes | 3.9 | No/no | 0.35 | 13.8 |
| 8 | F | 61 | Yes | 3.1 | No/no | 0.08 | 2.0 |
| 9 | M | 45 | Yes | 4.0 | Yes/no | 0.28 | 17.2 |
| 10 | M | 69 | Yes | 3.3 | Yes/no | 0.46 | 16.2 |
| 11 | F | 52 | Yes | 2.1 | Yes/no | 0.43 | 27.5 |
| 12 | M | 51 | Yes | 3.6 | Yes/yes | 0.38 | 15.7 |
| 13 | F | 66 | Yes | 2.8 | Yes/no | 0.27 | 9.2 |
| 14 | F | 74 | Yes | 4.5 | Yes/no | 0.20 | 8.0 |
| 15 | F | 25 | No | 4.8 | Yes/yes | 0.15 | 4.5 |
| 16 | F | 62 | Yes | 4.2 | Yes/yes | 0.30 | 6.8 |
| 17 | F | 71 | No | 6.0 | Yes/no | 0.39 | 20.3 |
| 18 | F | 65 | Yes | 4.5 | Yes/yes | 0.28 | 10.2 |
| 19 | F | 58 | Yes | 5.2 | No/no | 0.17 | 8.0 |
| 20 | F | 46 | Yes | 2.1 | Yes/no | 0.37 | 16.2 |
| 21 | F | 48 | No | 5.0 | No/no | 0.11 | 3.0 |
| 22 | F | 68 | Yes | 1.9 | No/no | 0.21 | 7.0 |
| 23 | F | 24 | No | 4.2 | NA | 0.13 | 5.8 |
| 24 | F | 57 | No | 4.7 | No/no | 0.18 | 5.3 |
| 25 | M | 19 | Yes | 3.8 | Yes/yes | 0.09 | 4.2 |
| Of 25 RA patients | 68% F, 32% M | median 61 (19–75) | 76% seropositive | Median 4.0 (1.9–6.0) | | Median 0.28 (0.08–0.63) | Median 10.2 (2.0–51.0) |
| Of all RA patients | 80% F, 20% M | median 55 (19–82) | 79% seropositive | Median 4.0 (1.1–7.1) | | Median 0.20 (0.033–0.63) | Median 6.7 (0.8–51.0) |
| HC1 | F | 58 | NA | NA | Yes/no | 0.38 | 11.8 |
| HC2 | M | 61 | NA | NA | Yes/yes | 0.09 | 0.8 |
| HC3 | F | 50 | NA | NA | No/no | 0.36 | 26.0 |
| HC4 | M | 58 | NA | NA | No/no | 0.26 | 5.4 |
| HC5 | F | 65 | NA | NA | Yes/no | 0.35 | 24.7 |
| HC6 | F | 50 | NA | NA | No/no | 0.24 | 4.1 |
| HC7 | F | 62 | NA | NA | Yes/no | 0.23 | 4.4 |
| HC8 | F | 44 | NA | NA | Yes/no | 0.12 | 2.7 |
| HC9 | M | 55 | NA | NA | No/no | 0.18 | 3.0 |
| HC10 | M | 52 | NA | NA | Yes/yes | 0.08 | 1.7 |
| HC11 | M | 48 | NA | NA | No/no | 0.04 | 0.6 |
| HC12 | F | 57 | NA | NA | Yes/no | 0.15 | 6.2 |
| HC13 | F | 56 | NA | NA | Yes/no | 0.13 | 4.6 |
| HC14 | M | 70 | NA | NA | Yes/no | 0.17 | 2.4 |
| HC15 | F | 47 | NA | NA | No/no | 0.30 | 7.1 |
| HC16 | M | 48 | NA | NA | Yes/no | 0.18 | 11.1 |
| HC17 | F | 21 | NA | NA | Yes/no | 0.08 | 7.1 |
| HC18 | M | 60 | NA | NA | No/no | 0.38 | 12.0 |
| HC19 | M | 43 | NA | NA | Yes/no | 0.38 | 17.5 |
| HC20 | F | 66 | NA | NA | Yes/no | 0.26 | 7.1 |
| Of HC | 55% F, 45% M | Median 56 (21–70) | | | | Median 0.20 (0.043–0.39) | Median 5.8 (0.6–26.0) |

DAS28, disease activity index in 28 joints; F, female; M, male; NA, not available; SE, shared epitope.
Immunogene panel sequencing was performed on both CD4+ and CD8+ cells of 25 newly diagnosed RA patients and 20 healthy controls (HCs), and the clinical characteristics are presented in the table. More detailed clinical information is presented in Supplementary Table 2. Summarized data are presented for all 82 newly diagnosed RA patients, for the 25 patients sequenced with the immunogene panel, and for the healthy controls. The results are presented as median values (range in parenthesis), if applicable. HLA-DRB1 alleles that predispose to seropositive RA, namely, HLA-DRB1*01, HLA-DRB1*04, HLA-DRB1*10 and HLA-DRB1*14:02; Clonality, productive clonality index based on TCRB sequencing; Largest CD8+ clone, the largest productive T-cell clone size, shown as percentage of all CD8+). For the shared epitope, data are presented showing the shared epitope status of both alleles (yes/yes, both DRB1 alleles were SE; yes/no, only one DRB1 allele was SE; no/no, neither DRB1 allele was SE).

cells. After comparing CD8+ to CD4+, we repeated the opposite comparison, using CD8+ as a control sample.

Median target coverage for the panel was 397 for CD4+ and 386 for CD8+ cells in RA patients, and 465 and 437 in healthy controls, respectively (Supplementary Table 5). All putative candidate variants were confirmed with targeted amplicon sequencing, a method with a sensitivity of 0.5% (ref. 14). A list of the putative variants is included as Supplementary Data 1.

Amplicon-sequencing enabled us to confirm nine somatic variants occurring exclusively in CD8+ cells from four individual RA patients and one variant in one healthy control (Fig. 2a, Table 2, Supplementary Table 6 and Supplementary Fig. 8a–c).

To discover mutations in genes that were not included in the targeted immunogene panel, we performed exome sequencing on flow-sorted CD8+ lymphocytes from 3 RA patients. The subjects

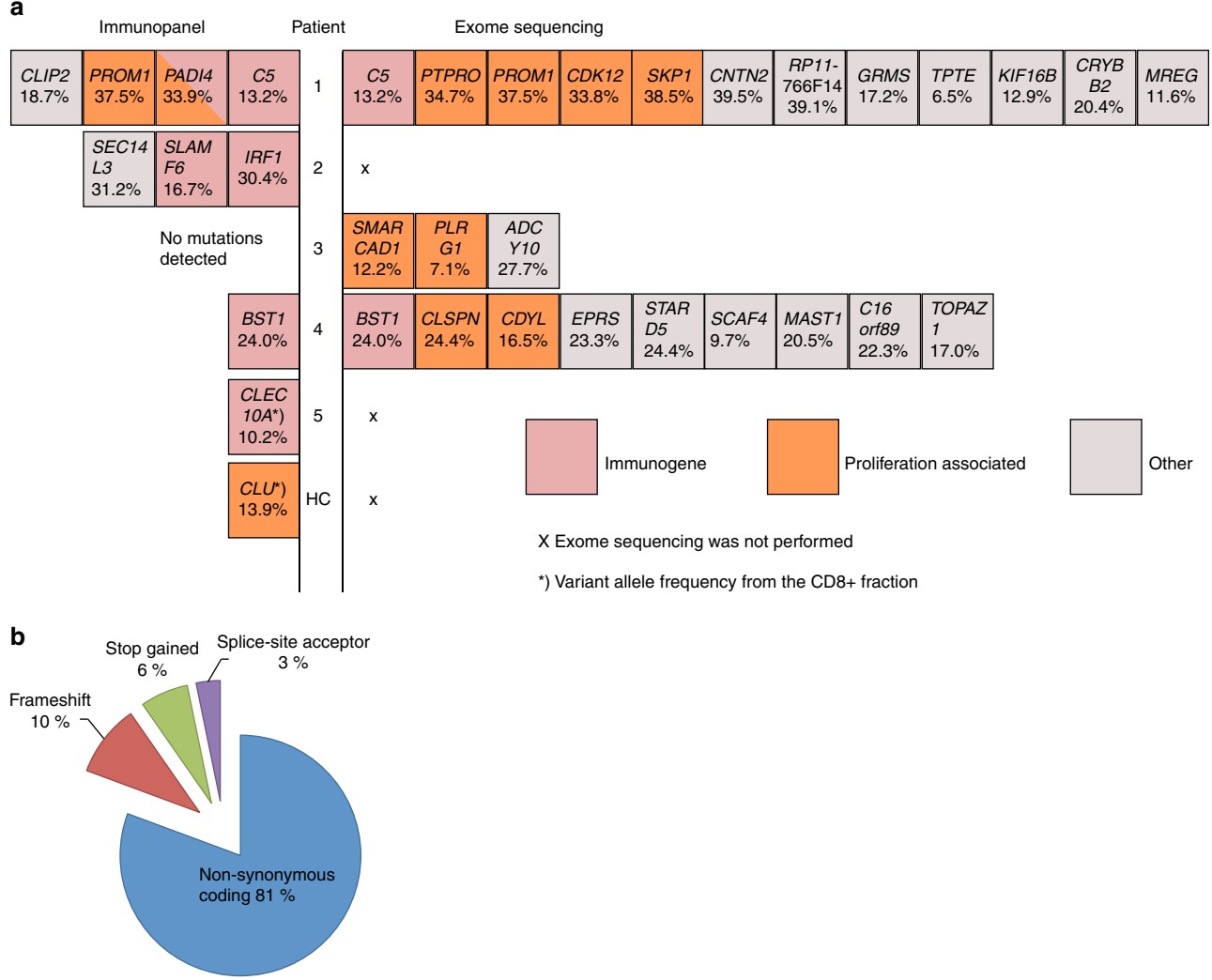

**Figure 2 | Somatic mutations are detected in 5 patients and in 1 healthy control.** (**a**) Somatic mutations were discovered in newly diagnosed RA patients and in one healthy (HC) by using the immunogene panel (presented on the left side), and exome sequencing (presented on the right side of the figure). Proliferation-associated and immune-related genes are highlighted with different colours. Percentages below the gene names denote the VAFs in CD8 + cells (immunogene panel) or expanded Vβ cells (exome). (**b**) The frequency of non-synonymous coding, nonsense-, frameshift-, and splice-site mutations shown as percentage of all identified mutations. Silent mutations were excluded from the bioinformatics analysis.

were selected on the basis of the initial flow-cytometry-based clonality screen, and they all harboured exceptionally large cell populations that were stained by a specific Vβ family antibody. A unique Vβ chain-specific antibody stained 44–55% of the total CD8 + lymphocyte pool, depending on the patient. The sequencing data from the sorted CD8 + Vβ-enriched lymphocyte pool were compared with those from the polyclonal CD4 + T-cell pool (germline control). Exome sequencing followed by confirmative amplicon sequencing revealed 24 novel somatic mutations (Fig. 2a, Supplementary Table 6, Supplementary Fig. 8d–f and Table 2) in the sorted CD8 + T-cell fractions. All putative variants are included in Supplementary Data 1. The exome-sequencing results were concordant with the immunogene panel sequencing regarding the genes that were included in the panel.

In total, we identified 30 novel somatic mutations in CD8 + cells of five RA patients. In contrast, only one healthy control harboured a single mutation in CD8 + cells. Immunopanel deep sequencing identified mainly immunity-associated genes, whereas exome sequencing added to the analysis by revealing somatic variants in proliferation-associated and unclassified genes (Fig. 2a).

The identified mutations were classified into non-synonymous (missense) coding- (81%), nonsense- (6%), frameshift- (10%) and splice-site acceptor mutations (3%; Fig. 2b). Silent mutations were excluded from the bioinformatics pipeline analysis. Table 2 reports scores predicting the effect of each mutation on the protein, on the basis of amino-acid conservation and structural information (SIFT; http://sift.jcvi.org/[16–20] and Polyphen-2; http://genetics.bwh.harvard.edu/pph2/[21]). Of the non-synonymous mutations, 32% (8/25) were predicted to be probably damaging by both Polyphen-2 and SIFT. All identified mutations were also queried against the Catalogue of somatic mutations in cancer (COSMIC, http://cancer.sanger.ac.uk/cosmic)[22] database, and four of the identified mutations have been reported in cancer (Supplementary Table 7).

**Somatic mutations are restricted to the expanded CD8 + clones.** Patient 1 was a 74-year-old male with palindromic rheumatism (a subtype of RA with sudden attacks of joint pain and swelling). At diagnosis, the flow-cytometry screen detected a CD8 + Vβ13.1 + T-cell population composing 50.4% of total CD8 + T

**Table 2 | Effects of the discovered mutations on the structure and function of affected proteins as determined by bioinformatic prediction tools.**

| Patient | Cell fraction | Ref | Var | Gene | Effect | Exon | Transcript | AA change | SIFT | Poly Phen | Var freq (%) |
|---|---|---|---|---|---|---|---|---|---|---|---|
| **Immunopanel** | | | | | | | | | | | |
| 1 | CD8+ | G | A | PADI4 | NS | 10 | ENST00000375448 | A359T | 0.01 | 0.91 | 17.5 |
| 1 | CD8+ | T | A | PROM1 | NS | 4 | ENST00000447510 | N344Y | 0.01 | 0.96 | 13.3 |
| 1 | CD8+ | A | T | C5 | NS | 9 | ENST00000223642 | C1532S | 0.01 | 0.97 | 6.6 |
| 1 | CD8+ | *) | G | CLIP2 | SSA | - | ENST00000223398 | – | NA | NA | 5.5 |
| 2 | CD8+ | C | T | IRF1 | NS | 8 | ENST00000245414 | G231E | 0.29 | 0.04 | 6.5 |
| 2 | CD8+ | C | T | SEC14L3 | STOP | 4 | ENST00000215812 | W74* | NA | NA | 6.4 |
| 2 | CD8+ | A | C | SLAMF6 | NS | 4 | ENST00000368057 | F238C | 0.21 | 0.00 | 5.0 |
| 4 | CD8+ | GT | G | BST1 | FS | 7 | ENST00000382346 | F220fs | NA | NA | 8.3 |
| 5 | CD8+ | C | T | CLEC10A | NS | 8 | ENST00000254868 | A235T | 0.23 | 0.08 | 10.2 |
| HC5 | CD8+ | T | C | CLU | NS | 1 | ENST00000560366 | H26R | **) | **) | 13.9 |
| **Exome sequencing** | | | | | | | | | | | |
| 1 | CD8+Vβ13.1+ | C | T | RP11-766F14.2.1 | NS | 1 | ENST00000511828 | R1077H | 0.01 | 0.99 | 35.5 |
| 1 | CD8+Vβ13.1+ | A | T | PTPRO | NS | 11 | ENST00000281171 | M665L | 0.18 | 0.17 | 32.4 |
| 1 | CD8+Vβ13.1+ | T | A | PROM1 | NS | 9 | ENST00000447510 | N344Y | 0.04 | 0.96 | 28.0 |
| 1 | CD8+Vβ13.1+ | G | A | CNTN2 | NS | 17 | ENST00000331830 | G725R | 0.33 | 0.51 | 28.0 |
| 1 | CD8+Vβ13.1+ | A | AG | SKP1 | FS | 5 | ENST00000517625 | E73fs | NA | NA | 25.0 |
| 1 | CD8+Vβ13.1+ | C | T | GRM5 | NS | 4 | ENST00000418177 | V313M | 0.00 | 1.00 | 21.3 |
| 1 | CD8+Vβ13.1+ | C | T | CRYBB2 | NS | 6 | ENST00000398215 | R160C | 0.01 | 0.38 | 18.0 |
| 1 | CD8+Vβ13.1+ | A | T | MREG | NS | 5 | ENST00000236976 | F183I | 0.12 | 0.60 | 15.8 |
| 1 | CD8+Vβ13.1+ | G | A | CDK12 | NS | 2 | ENST00000447079 | G499E | 0.40 | 0.35 | 13.0 |
| 1 | CD8+Vβ13.1+ | A | T | C5 | NS | 38 | ENST00000223642 | C1532S | 0.01 | 0.97 | 10.6 |
| 1 | CD8+Vβ13.1+ | C | G | TPTE | NS | 23 | ENST00000361285 | E504Q | 1.00 | 0.03 | 10.2 |
| 1 | CD8+Vβ13.1+ | C | T | KIF16B | NS | 19 | ENST00000408042 | E689K | 0.06 | 0.86 | 9.3 |
| 1 | CD8+Vβ13.1 | G | A | PADI4***) | NS | 10 | ENST00000375448 | A359T | 0.01 | 0.91 | 20.0 |
| 3 | CD8+Vβ7.2+ | C | A | ADCY10 | NS | 23 | ENST00000367851 | M1103I | 0.19 | 0.15 | 24.8 |
| 3 | CD8+Vβ7.2+ | T | G | SMARCAD1 | NS | 20 | ENST00000359052 | L851W | 0.00 | 0.98 | 12.9 |
| 3 | CD8+Vβ7.2+ | T | C | PLRG1 | NS | 3 | ENST00000393905 | Q71R | 0.57 | 0.00 | 7.2 |
| 4 | CD8+Vβ1+ | T | C | SCAF4 | NS | 20 | ENST00000286835 | Q966R | 0.61 | 0.00 | 31.8 |
| 4 | CD8+Vβ1+ | GT | G | BST1 | FS | 7 | ENST00000382346 | F220fs | NA | NA | 30.6 |
| 4 | CD8+Vβ1+ | G | A | STARD5 | STOP | 3 | ENST00000302824 | Q90* | NA | NA | 27.6 |
| 4 | CD8+Vβ1+ | C | T | CLSPN | NS | 20 | ENST00000318121 | G1131S | 0.00 | 1.00 | 22.2 |
| 4 | CD8+Vβ1+ | C | T | C16orf89 | NS | 1 | ENST00000350219 | R83K | 0.64 | 0.07 | 19.5 |
| 4 | CD8+Vβ1+ | T | C | EPRS | NS | 20 | ENST00000366923 | N982S | 0.58 | 0.00 | 19.2 |
| 4 | CD8+Vβ1+ | TA | T | TOPAZ1 | FS | 2 | ENST00000309765 | N628fs | NA | NA | 14.5 |
| 4 | CD8+Vβ1+ | C | T | MAST1 | NS | 13 | ENST00000251472 | R496C | 0.01 | 1.00 | 10.3 |
| 4 | CD8+Vβ1+ | C | G | CDYL | NS | 4 | ENST00000328908 | A190G | 0.34 | 0.41 | 8.1 |

AA, amino acid; FS, frameshift; NS, non-synonymous; ref, reference base; SIFT, SIFT score for the non-synonymous mutation (0-deleterious, 1-tolerated); SSA, splice-site acceptor; STOP, stop codon introduced; Polyphen, polyphen-2 score for the non-synonymous mutation (0-benign, 1-damaging); var, variant base.
CD4+ and CD8+ cells from 23 patients and 7 healthy controls were sequenced with the immunogene panel of 986 genes, and 3 patients' sorted CD8+ Vβ clones underwent exome sequencing. Immunogene panel sequencing detected mutations in 4 patients and 1 healthy control, whereas exome sequencing detected several mutations in each expanded clone. Polyphen and SIFT scores were determined for the transcript with the most deleterious prediction value. SIFT scores <0.05 and Polyphen-2 scores >0.85 were considered probably damaging (shaded with grey). . *) 5′-GGCTGACCCCAGC-3′ **) Prediction algorithms did not return scores because the mutation resided outside the consensus coding sequence set. ***) PADI4 mutation site coverage was low (10–12 ×) in exome sequencing and thus did not permit high-confidence mutation detection.

cells (Supplementary Fig. 9). TCRB deep sequencing revealed two large (34% and 31%) T-cell clones (Fig. 3a). These two clones remained stable during follow-up despite immunosuppressive therapy with methotrexate, hydroxychloroquine and sulfasalazine (Fig. 3b). Immunogene panel sequencing revealed mutations in the PADI4, PROM1, C5, and CLIP2 genes, and exome-sequencing additional mutations in 10 genes (Fig. 2a). Interestingly, one nonsense mutation was detected in the PTPRO gene, which is known to downregulate STAT3 activity, and phosphoflow analysis indicated an exceptionally high amount (20–29%) of phospho-STAT3-positive CD8+ T cells in this patient (Supplementary Fig. 11). Amplicon sequencing of sorted CD8+ Vβ13.1+ and CD8+ T cells not expressing Vβ13.1 confirmed that the mutations existed exclusively in the expanded CD8+ Vβ13.1+ population (Fig. 3c).

Patient 2 was a 72-year-old female who also had palindromic rheumatism and a previous history of other inflammatory disorders: asthma, lichen ruber and atrophic gastritis. At the time of RA diagnosis, flow cytometric screening identified two unusually large populations of CD8+ T cells: Vβ1+ (14%) and Vβ13.6+ (11%) (Supplementary Fig. 9). In deep TCRB sequencing, the corresponding clones composed 4.6% and 7.8% of CD8+ cells. The sums of specific V–J recombinations are presented for CD8+ (Fig. 3d) and for sorted Vβ1+-stained fractions (Fig. 3e). These clones remained stable during immunosuppressive treatment (methotrexate, hydroxychloroquine, sulfasalazine) (Fig. 3f). Immunogene panel sequencing identified three mutations (IRF1, SLAMF6, SEC14L3) in the patient's CD8+ cells (Fig. 2a). Cell sorting and amplicon sequencing from a follow-up sample confirmed that the discovered mutations were restricted to the CD8+Vβ1+ population (Fig. 3g). In heterozygous mutations, the mutation's VAF is expected to be half of the clone size if the mutation occurs in all of the clone's cells. The VAFs of IRF1 and SEC14L3 mutations (30–32%) in the sorted CD8+Vβ1+ cells matched well with the clone size: TCRB sequencing confirmed that the flow-sorted CD8+Vβ1+ cells harboured a monoclonal 73% TRBV09-01 expansion, as defined by a unique nucleotide sequence.

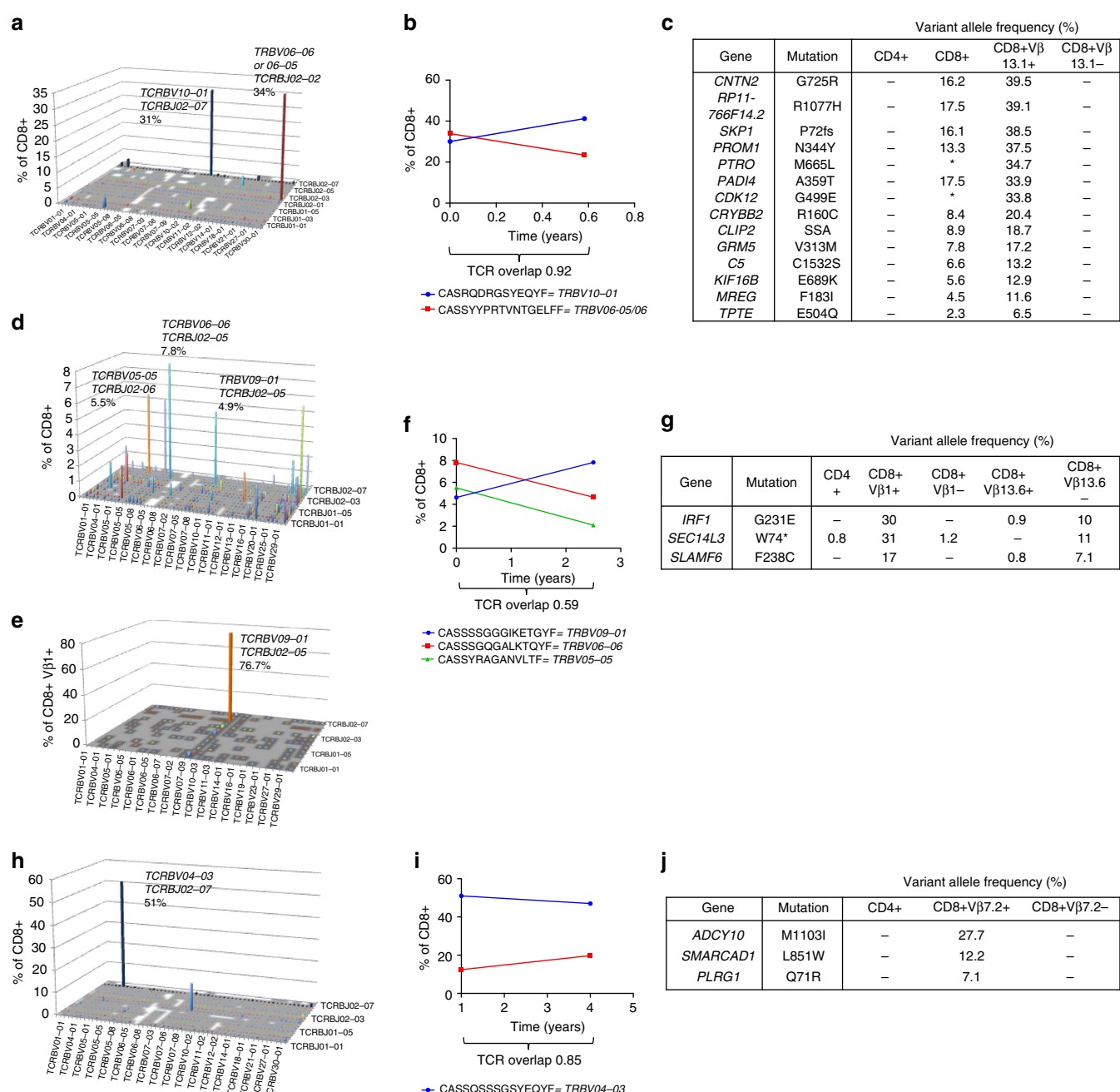

**Figure 3 | Somatic mutations occur in expanded CD8+ T-cell clones that persist during follow-up.** (**a**) The figure shows the CD8+ clonal architecture of patient 1, on the basis of TCRB sequencing and V-J genes. The Vβ13.1+ population detected by flow cytometry corresponded to clones using TCRBV06-05, 06-06, or 06-09 genes. (**b**) TCRB sequencing showed that patient 1 harboured two very large CD8+ T-cell clones. The two expanded clones of patient 1 persisted at a similar level during follow-up. The amino-acid TCRB sequences and the V genes of these unique clones are shown in the figure. (**c**) Amplicon sequencing of FACS-sorted cell fractions confirmed the identified mutations. The table presents the VAFs in each cell fraction. (**d**) The clonal architecture of patient 2's CD8+ pool as shown via V and J genes. Clones using TCRBV09-01 correspond to Vβ1+ antibody in flow cytometry. (**e**) Similarly presented TCRB sequencing results of flow-sorted Vβ1+ cells. When examining unique clones ('unique' defined by a unique nucleotide sequence) the largest clone composed 73% of all CD8+ Vβ1+ cells. (**f**) Patient 2 harboured several unique CD8+ T-cell clones at diagnosis. The TCRBV09-01 clone, in which mutations were observed, increased slightly during the follow-up. Amino-acid sequences derived from these unique clones are shown. The clone using TCRBV09-01 in this panel had the exact same nucleotide TCRB sequence as the largest clone in sorted Vβ1+ cells. (**g**) Amplicon sequencing results on FACS-sorted cells show the VAFs in each cell fraction. The low (<1%) VAFs found in CD4+ and Vβ13.6+ cells are considered sorting impurities, and thus the mutations in patient 2 occur exclusively in CD8+Vβ1+ cells. The mutation VAFs in CD8+Vβ1+ cells correspond well with the TCRBV09-01 clone size in sorted cells (**h**) The clonal landscape of CD8+ cells from patient 3. Clones using TCRBV04-03 gene correspond to Vβ7.2 usage in flow cytometry. (**i**) Patient 3 harboured a very large CD8+ T-cell clone at diagnosis, which persisted at a similar level during follow-up. (**j**) Amplicon sequencing of flow-sorted cell populations showed that the mutations detected in exome sequencing exist only in Vβ7.2 CD8+ T cells and not in other T cells.

Patient 3 was a 44-year-old male with seropositive erosive RA. Flow cytometry identified a large CD8 + Vβ7.2 + population composing 55% of all CD8 + T cells (Supplementary Fig. 9). TCRB sequencing revealed a CD8 + T-cell clone at 51% that corresponded to the flow cytometry result (Fig. 3h), and the clone remained stable during the follow-up (Fig. 3i) Immunogene panel sequencing did not reveal any mutations, but the mutations identified by exome sequencing in proliferation-associated genes (SMARCAD1, PLRG1, ADCY10) were restricted to the CD8 + Vβ7.2 + population (Fig. 3j).

Patient 4 was a 74-year-old female with seronegative disease. Flow cytometry analysis showed a CD8 + Vβ1 population at 44%, and the population harboured mutations in nine different genes (full list of affected genes in Fig. 2a, Table 2, and Supplementary Table 6). Patient 5 was a 59-year-old woman with seropositive RA, asthma and non-compaction cardiomyopathy. Flow cytometry did not detect large CD8 + T-cell populations expressing a certain Vβ; however, deep TCRB sequencing detected a clone composing 29% of all CD8 + cells (Table 1). Her CD8 + cells harboured one mutation in CLEC10A gene at 10% VAF (Fig. 2a).

In addition, one healthy control (HC5, a 65-year-old female) harboured a somatic mutation in CD8 + cells at a VAF of 13.9% (CLU, Fig. 2a). Deep TCRB sequencing revealed that the largest CD8 + clone composed 25% of CD8 +. No other somatic mutations were identified in the healthy controls despite the higher median target coverage in healthy controls compared with that in RA patients (Supplementary Table 5).

**RNA sequencing confirms the somatic mutations**. RNA sequencing was performed to assess the expression levels of the mutated genes both in the expanded Vβ population and in polyclonal CD8 + cells from the same patient (patients 1–3, Fig. 4a). All somatic mutations with sufficiently high expression were confirmed from the RNA sequencing data (Fig. 4a).

Paired analysis of Vβ-antibody-positive and -negative fractions revealed a proliferation and survival associated signature in the expanded Vβ populations (Fig. 4b). Functional annotation using RefSeq summaries indicated a shared functional state that supports survival and enhanced proliferation in the cell fractions containing the expanded clones. When each mutation-harbouring Vβ-stained cell population was analysed alone, several immunologically relevant characteristics were assigned to each mutation-bearing fraction (Fig. 4b). Interestingly, the PADI4 mutation resulted in dramatic downregulation of mRNA transcripts in patient 1 with the A359T mutation in his CD8 + Vβ13.1 + cell population.

**The mutated clones show an effector-memory phenotype**. We studied the phenotype of the mutation-harbouring T-cell populations with multicolour-flow cytometry, using follow-up samples collected during ongoing antirheumatic therapy. In patients 1 and 3, the CD8 + populations expressing the specific Vβ (13.1 and 7.2, respectively) showed an effector-memory (CCR7-CD45RA-) phenotype, whereas the polyclonal CD8 + cells not expressing the same Vβ and CD4 + cells showed a balanced T-cell distribution (Fig. 5a,b). In patient 2, the CD8 + Vβ1 + population had a terminally differentiated effector-memory phenotype (CCR7-CD45RA +; Fig. 5c).

Cytomegalovirus (CMV)-reactive T-cell clones commonly occur in elderly individuals and are related to the increased clonality of CD8 + T cells[23]. Thus, we compared all RA CD8 + clones exceeding 1% against published viral-specific TCRB sequences (listed in the Supplementary Information), but only a few TCRs have previously been reported to be virus specific (Supplementary Table 8). The TCRB sequences of major CD8 + clones in patients harbouring mutated cells were not among the published viral-specific TCRB sequences. Additionally, we tested CMV reactivity in patients 1 and 2 with a NLV-peptide-specific pentamer, and only a minor proportion of CD8 + T cells were CMV NLV-epitope specific (Supplementary Fig. 10).

**The confirmed mutations are not reproduced in other RA patients**. The somatic mutations identified by the immunogene panel in patients 1 and 2 (PADI4, PROM1, C5, IRF1, SLAMF6 and SEC14L3) were screened in 82 RA patients with targeted amplicon sequencing (coverage 4,000–79,000 ×, sensitivity 0.5% (ref. 14)). For most patients, both CD8 + and CD4 +-enriched fractions were used in the analysis. No coding region mutations were detected within the amplicons covering the mutation sites. Finally, because some RA patients showed marked STAT3 phosphorylation in phosphoflow analyses (Supplementary Fig. 11), we examined all 82 patients for previously described somatic variants in exon 21 of STAT3. Amplicon sequencing coverage reached 4,800–110,000, but no mutations in STAT3 were identified. In conclusion, none of the screened mutations occurred in other patients.

**Discussion**

This study presents the novel discovery of somatic mutations in both immune- and proliferation-associated genes in the CD8 + T cells of patients with newly diagnosed, untreated RA. Several recent reports have shown that somatic mutations originating from haematopoietic precursor cells occur in blood cells of the healthy elderly (clonal haematopoiesis) and that the prevalence of clonal haematopoiesis increases with age. Many of these mutations occur recurrently in myeloid malignancies, and their presence in healthy individuals increases the risk of haematological cancers[3–6]. Interestingly, somatic clonal mosaicism in blood- and buccal cells has also been shown to associate with solid tumours[24]. To underscore the developmentally late emergence of the mutations in CD8 + effector T cells, we used CD4 + cells as the comparator in our study. This method excluded mutations that had already arisen in precursor cells. To our knowledge, this is the first study showing the emergence of somatic mutations in effector T cells in patients with a non-malignant autoimmune process.

Somatic mutations were identified in genes that were expressed in CD8 + lymphocytes and that are known regulators of immunity (such as IRF1 and SLAMF6) and cell proliferation (PLRG1). Notably, no mutations were detected in CD4 + cells, and only one somatic mutation was identified in 1 out of the 20 healthy controls.

In 16% of the cases, the mutations were either frameshift- or nonsense-mutations, and thus functional consequences were apparent. To evaluate the effects of the identified non-synonymous mutations on the structure and function of the affected proteins, we used two separate bioinformatics tools (Polyphen-2 and SIFT), and both indicated likely deleterious effects in 32% of the non-synonymous cases. RNA-sequencing results demonstrated that many of the mutated genes were highly expressed in CD8 + cells, and thus mutations in these genes have the capacity to modulate cellular functionality. Interestingly, the RNA sequencing data also revealed a survival/proliferation-associated signature in the flow-sorted cells in which the mutations were located. These data together suggested that the expanded fractions differ from the highly polyclonal background pool. The differences may be both related and unrelated to the identified somatic mutations. More research is warranted to

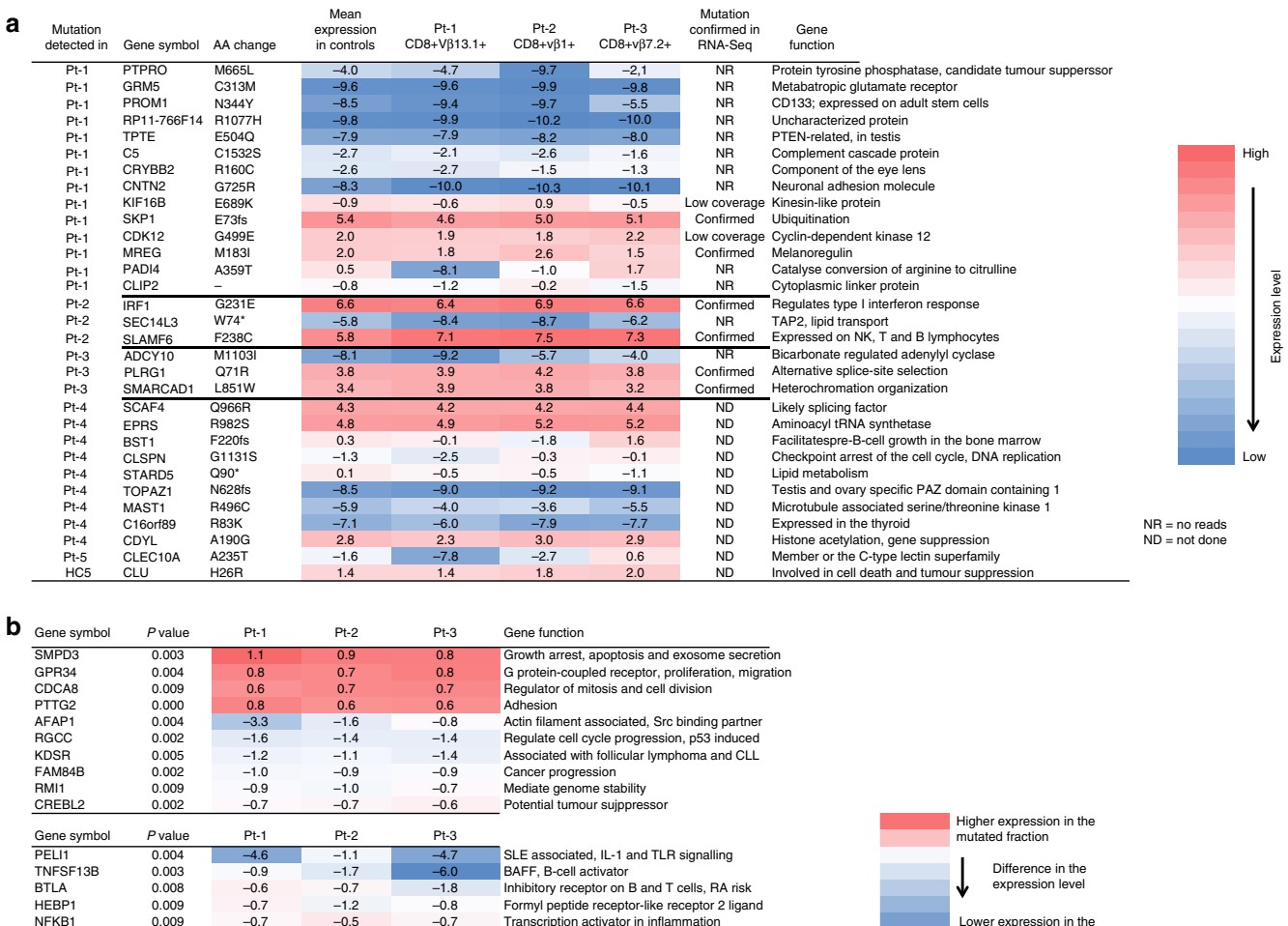

**Figure 4 | Expression levels of mutated genes and a survival signature in mutation-harbouring cells.** CD8 + cells from 3 healthy controls as well as mutation-harbouring CD8 + Vβ + and polyclonal CD8 + T cells from patients 1, 2, and 3 were sorted with flow cytometry. (**a**) RNA sequencing was performed on the sorted fractions. The normalized mean expression levels calculated from the healthy CD8 + T cells and from the patients' polyclonal CD8 + fractions are presented in the control expression column. In the following columns, each sorted population containing the mutated clone is presented individually. (**b**) The upper panel presents proliferation/survival -associated genes identified by paired analysis of the mutation-harbouring cell populations and their polyclonal background populations. Immunologically important transcripts are presented in the lower panel. The transcripts were filtered with threshold expression level 0, FC > 1,5 or < − 1,5 and P < 0.01. Abbreviations: Aa, amino acid; fs, frameshift; *, stop-codon gained; -, splice-site acceptor mutation. The function of the gene was retrieved from the RefSeq database and noll-hypothesis was tested using ANOVA in the Qlucore software package.

understand in detail the biological consequences of the identified somatic mutations.

Previous studies with smaller cohorts have reported clonal CD8 + T-cell expansions in RA patients' blood samples[25–27]. In contrast to these studies, we studied the clonal architecture of CD8 + T cells by NGS TCRB sequencing. TCRB sequencing identified large clonal expansions in both RA patients as well as in healthy controls, and no difference in clonality was found between the two groups. This result was not unexpected because large CD8 + T-cell clones are not uncommon in the elderly[28,29], and the median ages of our patient and control groups were essentially identical. Further, our results showed that age was positively correlated with the clonality index (older individuals harboured more large clones than younger people), a result in line with findings from studies showing that age is associated with decreased TCR diversity[30].

CD8 + T cells display more clonal characteristics than CD4 + cells do (as shown by our flow cytometry data, and sequenced

by Qi *et al.*[30]). Vigorous proliferation involving repeated DNA-replication cycles predisposes the clone to an increased risk of mutagenesis. Larger cell clones also enable detection of mutations with larger VAFs, thus technically facilitating identification of somatic variants. Despite the similar CD8 + clonality in the RA patients and controls, somatic mutations were observed in 5 (out of 25) RA patients and only in 1 (out of 20) healthy control. We conclude that the chronic inflammatory state in RA[9] may facilitate the acquisition of somatic mutations, and thus, we hypothesize that somatic mutations may also occur in other inflammatory conditions.

Possible TCR target antigens in RA have been studied intensively. Whereas CD4 + cells have received much attention, CD8 + cells have mainly been studied as a potential link between autoimmunity and viral infections. Virus-specific (Epstein-Barr virus (EBV), CMV, influenza) CD8 + T-cell enrichment has been reported in RA patients' synovial fluid[31], and EBV has been reported as a possible target antigen for expanded T cells in RA

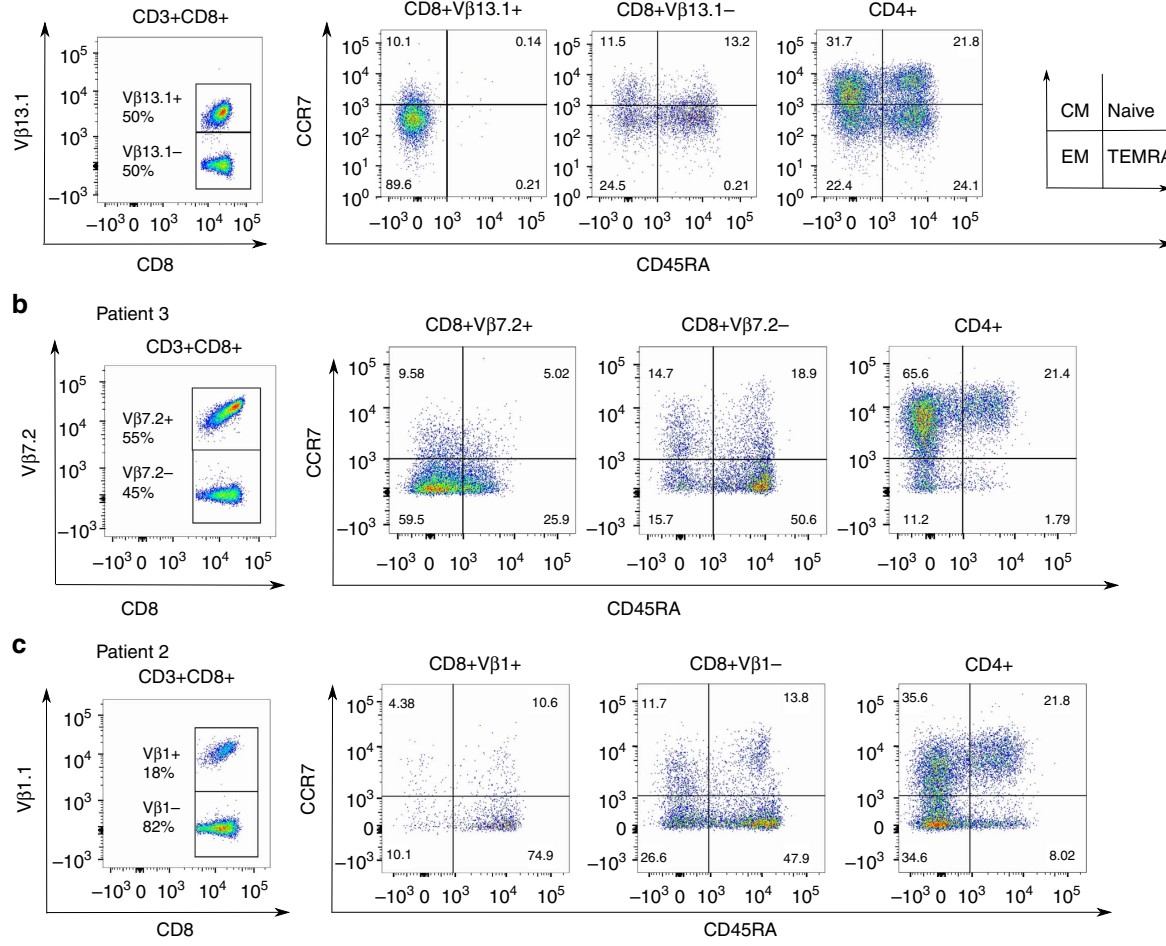

**Figure 5 | Expanded CD8+ T-cell clones are effector-memory T cells.** Multicolour-flow cytometry was used to determine the phenotype of patients' T cells in samples taken during the follow-up. CCR7 and CD45RA were used to determine whether the cells were central memory (CM), naive, effector-memory (EM) or terminal effector-memory RA-positive (TEMRA) cells. (**a**) The CD8+ Vβ13.1+ cells of patient 1 comprise 50% of all CD8+ T cells. The phenotype of CD8+Vβ13.1+, other CD8+, and CD4+ T cells are displayed in separate plots. (**b**) The phenotype of CD8+Vβ7.2+, other CD8+, and CD4+ T cells of patient 3 are displayed in separate plots. (**c**) The phenotype of expanded CD8+Vβ1+, other CD8+, and CD4+ T cells of patient 2 are presented in separate plots.

patients[32]. Virus-specific clones may display cross-reactivity towards autoantigens[33], thus enabling interaction with the autoimmune inflammation process. Because the antigen target of the expanded clones in our study is unknown the interpretation of the results is limited. None of the largest (>5%) CD8+ clones were among the previously published virus-specific TCRB sequences, but this approach does not rule out the possibility that the clones might be virus-specific. Future studies are needed to assess whether common viral- or auto-antigens are targets of these clones. However, TCRs show remarkable binding plasticity and cross-reactivity between allo-antigens and auto-antigens[34,35], thus enabling autoreactive properties independent of their primary antigen target.

We identified somatic mutations in memory-type[36] CD8+ T cells. In two patients, the mutations were discovered in clones without CD45RA expression (effector-memory cells), and in one patient the mutated clone was CD45RA-positive (terminally differentiated effector-memory phenotype cells). Both phenotypes represent cells that have encountered their antigen, proliferated, and participated in a cytotoxic response. After the antigen encounter, the clones failed to shrink and persisted in these patients. We hypothesize that this finding may be due, in part, to the somatic mutations that the cells have acquired during the proliferation process (Fig. 6).

Future studies are needed to elucidate whether the mutation-harbouring CD8+ T-cell clones truly participate in the pathogenesis of an autoimmune disease, or whether they are like 'scars' in the genome after vigorous antigen-driven proliferation. Even though the role of CD8+ cells in RA remains less well-established than the role of CD4+ T cells, there is a growing body of evidence supporting the role of CD8+ T cells as regulators of autoimmune arthritis[37]. Interestingly, the genes found to be mutated in our patient material (such as SLAMF6 (ref. 38), PADI4 (ref. 39) and IRF1 (ref. 40)) have been shown to regulate autoimmunity and to be associated with the initiation and progression of RA (PADI4 and IRF1).

Our study highlights novel putative mechanistic similarities between autoimmunity and cancer: somatic mutations in disease-associated cells. The occurrence of mutations in terminally differentiated T cells may discriminate a benign disorder from a malignant transformation (such as leukaemia or lymphoma), because terminally differentiated effector-memory cells have limited proliferation potential. These data lay the groundwork for future studies seeking to understand the role of somatic

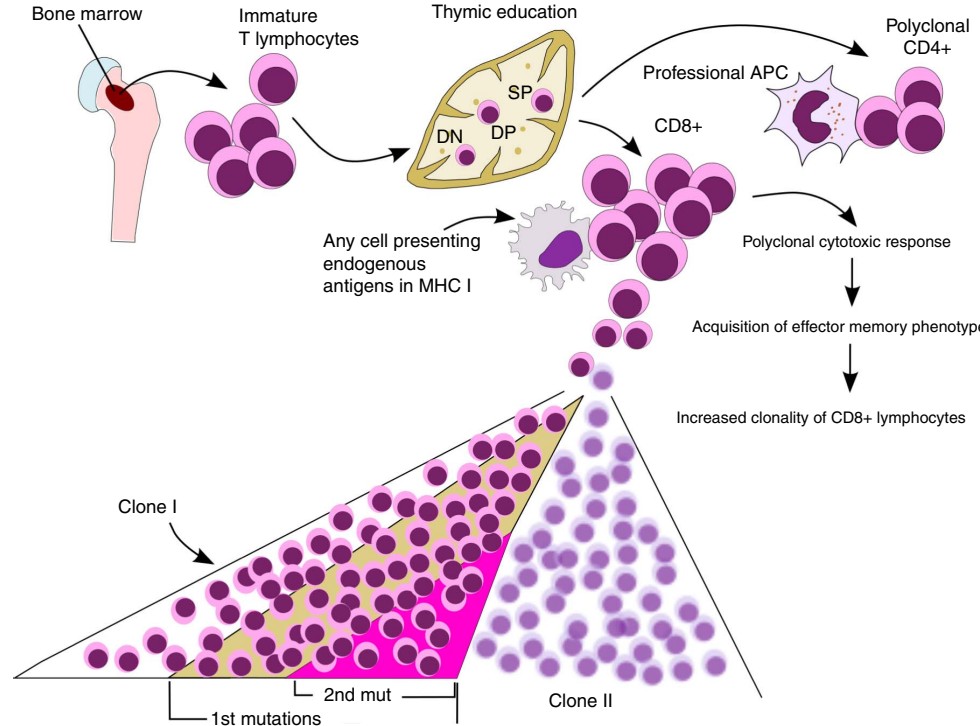

**Figure 6 | Hypothesis for the accumulation of somatic mutations.** Mature T cells undergo thymic selection, and during the expansion phase after antigen stimulation, mutation(s) occur (1st mutation). When the cell population continues to grow, additional mutations may arise in the same clone, thereby resulting in different VAFs in the same clonal population. In our patient cohort, CD4 + cells presented a more equally distributed T-cell repertoire than did CD8 + T cells (analysed with flow cytometry), and we did not find any somatic variants in CD4 + cells. APC, antigen-presenting cell; DN, double negative; DP, double positive; SP, single positive; MHC, major histocompatibility complex.

mutations in autoimmune diseases and other non-malignant disorders.

## Methods

**Study design.** This is a prospective study of newly diagnosed RA patients approved by the Helsinki University Hospital Ethics Committee. Patients were enrolled to the study in the Department of Rheumatology (Helsinki University Hospital, Finland) and all newly diagnosed patients referred to our clinic between 2012–2013 fulfilling the ACR & EULAR 2010 classification criteria were aimed to be recruited. Written informed consents were required from all participants and the Declaration of Helsinki was followed.

**Patient samples.** Routine laboratory and clinical parameters were recorded from all participating patients. Clinical parameters included the total amount of tender and swollen joints, Disease Activity Score in 28 joints, Health Assessment Questionnaire and the patient's global assessment of disease activity on visual analogue scale, and the duration of symptoms before diagnosis. Peripheral-blood samples (30 ml EDTA blood) were obtained directly after diagnosis and before initiating treatment. For immunogene- and TCRB-sequencing, 20 healthy blood donor buffy coats (provided by the Histocompatibility Testing Laboratory, Finnish Red Cross Blood Service) served as controls.

**Sample preparation.** Peripheral-blood mononuclear cells (PBMCs) were separated from EDTA blood using Ficoll gradient separation (Ficoll-Paque PLUS, GE Healthcare, cat. no 17-1440-03). CD4 + and CD8 + cells were separated with magnetic bead sorting using positive selection for both fractions (AutoMACS, Miltenyi Biotec). Purities of the sorted fractions were confirmed with flow cytometry (FACSAria II, Becton Dickinson) and the gating strategy for all PBMC analysis is presented in Supplementary Fig. 12.

**Clonality analysis with flow cytometry.** Peripheral whole-blood samples were stained for lymphocyte clonality analysis with anti-CD3 (SK7, BD, cat. no 345767), anti-CD4 (SK3, BD, cat. no 345770), and anti-CD8 (SK-1, BD, cat. no 335822) and a panel of T-cell receptor β variable chain (TCR Vβ) antibodies (IOTest Beta Mark TCR V kit, Beckman Coulter Immunotech, cat. no IM3497), which recognize ∼70–80% of human TCR β V regions. All fluorochrome-conjugated antibodies were used according to the manufacturer's instructions. Similarly, all β variable chain (TCR Vβ) antibodies were used according to the manufacturer's instructions

and all antibody information is included in the kit user manual and also in the Supplementary Fig. 13. After staining, red blood cells were lysed with BD FACS Lysing Solution (Becton Dickinson Biosciences, cat. no 349202) and re-suspended to phosphate-buffered saline (PBS) with 2 mM EDTA. Samples were analysed with FACSAria II (Becton Dickinson) and FACSDiva software (Becton Dickinson). The gating strategy for all whole-blood stainings is presented in the Supplementary Fig. 13.

**Flow-assisted cell sorting.** Cell sorting was performed from PBMC fractions using FACSAria II (Becton Dickinson) or Sony SH800 (Sony Biotechnology Inc.). For sorting, PBMC cells were stained with anti-CD3 (SK7, BD, cat. no 345767 or 557851), anti-CD4 (SK3, BD, cat. no 345770), anti-CD8 (SK-1, BD, cat. no 335822 or 345772) and the appropriate anti-Vβ antibody. Purities of the sorted cell fractions were controlled with flow cytometry, and the purities were nearly 100%.

**Phenotyping and CMV-pentamer staining.** For phenotyping analyses, PBMCs were stained with anti-CD3 (SK7, BD, cat. no 345767), anti-CD4 (SK3, BD, cat. no 345770), anti-CD8 (SK1, BD, cat. no 335822), anti-CCR7 (150503, BD, cat. no 562555), anti-CD45RA (HI100, BD, cat. no 560673) and the appropriate Vβ antibody were used. For CMV-pentamer staining, cells were labelled with iTAg MHC pentamer (HLA-A-0201, NLVPMVATV, Proimmune, PE conjugate) and anti-CD3, CD8, and CD4. Samples were analysed with FACSVerse (Becton Dickinson) and FlowJo software.

**Phosphospecific flow cytometry.** For phosphospecific flow cytometry, 100 µl aliquots of whole-blood were supplemented with 5 µl anti-CD4 (FITC, SK3) and 3 µl anti-CD8 (SK1, BD). After a 15-min incubation at + 37 °C, fixation, red-cell lysis (1 × Lyse/Fix Buffer, BD) and permeablization followed (100 µl 1 × PhosFlow Perm/Wash Buffer I, BD). The phosphorylated intracellular targets were labelled using 5 µl anti-pSTAT3 (pTyr705)-PE (4/p-STAT3, BD), 5 µl anti-pSTAT3(pSer727)-AlexaFluor647 (49/p-STAT3, BD), 5 µl polyclonal rabbit anti-pJAK1(pTyr1022/1023, 1:10 dilution in PBS, Merck), 7 µl anti-pJAK2 (pTyr1007/1008, Santa Cruz Biotechnology), 7 µl anti-pJAK3(pTyr980, Santa Cruz Biotechnology) and 7 µl anti-SHP1(pTyr536, 1:10 dilution in PBS, Abcam). Anti-CD3 (SK7) was also added following permeabilization. F(ab')2 donkey anti-rabbit IgG (BD) was used as fluorescent secondary antibody in 40-min incubation as PE-conjugate for pJAK1 (2 µl), pJAK2 (5 µl) and pSHP1 (2 µl) and PerCP-conjugate for pJAK3 (2 µl). The samples were kept on ice and analysed

within 4 h with FACSCantoII flow cytometer and analysed with FACSDiva software (BD) as described previously[41,42]. Each patient sample was compared to a healthy control sample taken during the same week, which was used to determine gating for positive and negative populations: the histogram of a healthy control subject's sample was set not to cover more than 5% of the cells of the population, and this gate was copied to patient samples. The gating strategy for phosphorylation analysis is presented in the Supplementary Fig. 14.

**DNA extraction.** DNA was extracted from CD4- and CD8-enriched samples or from the PBMC fraction with Nucleospin Tissue DNA extraction kit (Machery Nagel, cat. no 740952.250) or Nucleospin Tissue XS kit (Machery Nagel, cat. no 740901.5) according to the manufacturer's instructions. DNA concentration was measured with Qubit 2.0 fluorometer (Life Technologies) using the Qubit HS or BR dsDNA Assay Kit (Life Technologies, Cat. Nos Q32854 and Q32853).

**TCRB CDR3 deep sequencing.** To confirm the clonality screening results obtained via flow cytometry, TCRB deep sequencing was performed from 65 patients, accompanied by sequencing of 20 healthy controls (whose samples also underwent immunogene panel sequencing). Genomic DNA was used in all cases. Sequencing and data analysis were conducted as previously described with ImmunoSEQ assay by Adaptive Biotechnologies Corp[43,44]. Only productive TCR sequences were included in all analyses in this report.

Clonality was calculated according to the formula:

$$\text{Clonality} = 1 - \frac{-\sum_{i=1}^{N} p_i \log_2(p_i)}{\log_2(N)} \quad (1)$$

where $p_i$ is the proportional abundance of the rearrangement $i$, and $N$ is the total number of rearrangements. The numerator of the equation is Shannon's entropy. TCR repertoire overlap between two samples was calculated with the following formula:

$$\text{Overlap} = \frac{\sum_{i}^{n} a_i + b_i}{A + B} \quad (2)$$

in which $a_i$ is the template count of clone $i$ in sample $A$, $b_i$ the template count of clone $i$ in sample $B$, $A$ the total number of templates in sample $A$, and $B$ the total number of templates in sample $B$.

**Immunopanel sequencing.** A list of 986 genes related to immunity and cancer was created (Supplementary Table 5), including genes in the annotated InnateDB database (http://www.innatedb.ca/). Additional potentially interesting targets such as JAK-STAT pathway-related genes were compiled from different publications[45–49]. The probes were designed based on the genomic coordinates for the coding parts and surrounding untranslated regions of the genes by the Nimblegen SeqCap EZ Developer system.

Two different DNA library preparation kits were used in this study: NEBNext (New England BioLabs, cat. no E6040L) and ThruPLEX DNA-seq (Rubicon Genomics, cat. no R400407). For the NEBNext kit, the DNA libraries were prepared from 1–3 μg DNA according to NEBNext DNA Sample Prep Master Mix Set 1 (New England BioLabs) manual with minor modifications: (1) 1 μl of the 25 μM adaptor was used in ligation instead of 10 μl of 15 μM adaptor. Illumina Index PE adaptor oligo mix (from primers 5′-GATCGGAAGAGCACACGTCT-3′ and 5′-ACACTCTTTCCCTACACGACGCTCTTCCGATCT-3′) was used in ligation. Both oligos were custom ordered from Sigma Aldrich (St Louis). (2) After each consecutive library preparation step the library was purified with Agencourt AMPure XP -beads (Beckman Coulter, cat. no A63881). (3) For each library 15 ng of ligated library was used for pre-capture PCR in five parallel reactions. Twelve amplification cycles were used. (4) The initialization step in the amplification was 2 min instead of 30 s and the denaturation step was 20 s instead of 30 s. (5) Illumina PCR Primer InPE 1.0 (5′-AATGATACGGCGACCACCGAGATCTACACTCTTT CCCTACACGACGCTCTTCCGATCT-3′) was used as a forward primer. For each library different, indexed, long reverse primer was used. Each indexed, long reverse primer was custom made and merged from the Illumina PCR Primer InPE 2.0 (5′-GTGACTGGAGTTCAGACGTGTGCTCTTCCGATCT-3′) and Illumina TruSeq Small RNA index primer (5′-CAAGCAGAAGACGGCATACGAGATNN NNNNGTGACTGGAGTTC-3′, NNNNNN being the selected small RNA index). The merged long reverse primer template was 5′-CAAGCAGAAGACGGCATAC GAGATNNNNNN GTGACTGGAGTTCAGACGTGTGCTCTTCCGATCT-3′ in which the NNNNNN sequence was replaced with selected index. Using these long reverse primers decreased the cycles needed in amplification from 19 to 12. All primers were custom ordered from Sigma Aldrich (5). Sufficient amount of library was produced, hence the LM-PCR step in the NimbleGen protocol was excluded. Library quantification was performed using 2100 Bioanalyzer High sensitivity kit (Agilent, Santa Clara, CA, USA, cat. no 5067-4626).

Target region captures were performed according to NimbleGen SeqCap EZ Exome Library SR User's Guide with minor modifications. NimbleGen captures were performed according to NimbleGen SeqCap EZ Exome Library SR User's Guide with the following exceptions. (1) An aliquot of 10 μl of 100 μM Illumina PCR Primer InPE 1.0, MPLEX_blockAdapter2.0 (5′-GTGACTGGAGTTCAGACG TGTGCTCTTCCGATCT-3′) and MPLEX_blockTail2.0 (5′-CAAGCAGAAGACG

GCATACGAGAT-3′) was used for blocking in each library. (2) An aliquot of 4 μl of the captured library was used in five parallel 50 μl amplification reactions. An aliquot of 1 μl of 100 μM NimbleGen PE-POST1 and PE-POST2 primers were used in each reaction. Twelve cycles were used.

For the ThruPLEX library preparation, 100 ng of gDNA of each was fragmented with Episonic Multi-Functional Bioprocessor 1100 (Epigentek Group Inc., Farmingdale, NY, USA) to mean fragment size of 300 bps,and 50 ng of fragmented DNA (fragmenting controlled with fragment analysis) was used in the following procedures. The libraries were processed according to ThruPLEX DNA-seq library preparation kit (Rubicon Genomics, Ann Arbor, MI, USA). Five amplification cycles were used. The libraries were quantitated with 2100 Bioanalyzer (Agilent Technologies, Santa Clara, CA, USA). The captures were performed according to Rubicon Genomics' protocol for Exome Capture of ThruPLEX Libraries with Roche NimbleGen SeqCap EZ Library (Rubicon Genomics, Ann Arbor, MI, USA). Roche Nimblegen custom capture probes were used to perform the capture. After hybridization the captures were performed according to NimbleGen SeqCap EZ Exome Library SR User's Guide.

All amplified libraries (pre-capture preparation with NEBNext or ThruPLEX) were purified with Agencourt AMPure XP beads (cat. no A63881) and quantified for sequencing using 2100 Bioanalyzer High sensitivity kit (cat. no 5067-4626). Sequencing was performed with the Illumina HiSeq system. Sequencing was done from both CD8 + and CD4 + cells of each patient.

**Bioinformatic analyses for somatic variant detection.** The principle of the somatic variant-calling pipeline has been previously described[13].

SeqPrep 0.4.5 was used to merge raw Illumina reads, and resulting paired reads were trimmed of B blocks in the quality scores from the end of the read. Trimmed reads < 36 basepairs were removed. Read-alignment was performed with Burrows–Wheeler Aligner version 0.6.2 (ref. 50) against the human genome GRCh37 reference-genome. Reads mapping to multiple genomic positions were removed. GATK Indel Realignment version 2.2–16 was used to refine the read alignment. After aligning, potential PCR duplicates were removed with Picard MarkDuplicates version 1.90.

VarScan2 somatic algorithm version 2.3.2 (ref. 51) was used to call high-confidence somatic mutations for each patient. The following parameters were used: strand-filter 1, min-coverage-normal 8, min-coverage-tumor 6, somatic $P$ value 1, normal-purity 1 and min-var-freq 0.05. Mutations were annotated using SnpEff version 4.0 (ref. 52) with the Ensembl v68 annotation database[53]. To filter out misclassified germline variants, the variants classified as common population variants in Single Nucleotide Polymorphisms database (dbSNP, build ID 130) were removed.

CD8 + cells included the expanded clones of interest, whereas CD4 + cells served as a germline control and was used to filter individual germline variants when the data were analysed. The bioinformatics pipeline was also run using CD8 + cells as germline control, but no somatic variants were confirmed in subsequent sequencing with the deep Amplicon method in CD4 + cells.

**Amplicon sequencing.** Targeted, deep amplicon sequencing (Illumina) is a PCR-based method with coverage up to over 100,000 × and a sensitivity of 0.5% VAF[14]. Amplicon sequencing was used to validate candidate mutations found in the immunopanel sequencing and to screen mutations from the whole patient cohort. Amplicon sequencing of selected genes was performed as described previously[14,54], and the primers used are listed in Supplementary Table 9.

The amplicon primers were designed to cover the exon that harboured the identified mutation. Either 1-step or 2-step PCR was used. The 1-step PCR was performed in a volume of 20 μl containing 10 ng of sample DNA, 10 μl of 2 × Phusion High-Fidelity PCR Master Mix (Thermo Fisher Scientific, cat. no F531L), 0.025 μM of each locus-specific primer, 0.5 μM of index primer 1 (P5) and 0.5 μM of index primer 2 (P7), supplemented with water to the final volume (Sigma-Aldrich). In this reaction, the locus-specific primers are present in limiting quantities. Adaptor primers amplify these products further. For the 2-step PCR protocol, the first PCR was done in a volume of 20 μl containing 10 ng of sample DNA, 10 μl of 2 × Phusion High-Fidelity PCR Master Mix and 0.375 μM of each locus-specific primer, supplemented with water to the final volume. The second PCR was done in a volume of 20 μl containing 1 μl of the amplified product from the first PCR, 10 μl of 2 × Phusion High-Fidelity PCR Master Mix, 0.375 μM of index primer 1 and 0.375 μM of index primer 2.

DNA Engine Tetrad 2 (Bio-Rad Laboratories) or G-Storm GS4 (Somerton) thermal cyclers were used to cycle the sample according to the program: initial denaturation at 98 °C 30 s, 30 cycles at 98 °C for 10 s, at 59–63 °C for 30 s, and at 72 °C for 15 s, and the final extension at 72 °C for 10 min. If a second PCR cycle was used (in 2-step PCR protocols), the second PCR was done according to the program: initial denaturation at 98 °C 30 s, eight cycles at 98 °C for 10 s, at 65 °C for 30 s, and at 72 °C for 20 s, and the final extension at 72 °C for 5 min.

The amplified samples were pooled together without exact quantification and the pool was purified with Agencourt AMPure XP beads (Beckman Coulter, cat. no A63881) twice using 0.8 × volume of beads compared to the sample pool volume, which removes effectively primer dimers from the sample pool. Agilent 2100 Bioanalyzer (Agilent Genomics) was used to quantify amplification performance and yield of the purified sample pools. Sample pools were sequenced with Illumina

MiSeq System using Illumina MiSeq Reagent Kit v2500 cycles kit (Illumina, cat. no MS-102-2003).

Sequencing reads were aligned to the human hg19 genome with Bowtie2. GATK IndelRealigner was used for local realignment near indels. Low-quality reads were also aligned to the genome, but quality scores were used to exclude error bases in further analysis. All variant alleles with count >5 and VAF of over 0.5% were called. A specific frequency ratio was calculated by dividing the ratio of variant calls/number of all the bases (at a position) by the ratio of variant allele quality sum/quality sum of all bases. This ratio was used to filter out false positive variants: variants with a frequency ratio <0.8 were excluded. In this study, the mutation was considered to be true if it was present in the sequenced sample with a similar VAF as reported by exome/immunogene panel sequencing, and if the variant was absent from the control sample from the same patient.

**Exome sequencing.** The expanded CD8 + Vβ clones of patients 1, 2 and 3 were sorted with FACSAria II, and underwent exome (50 ng of DNA) sequencing with Nextera Rapid Exome Kit (Illumina, cat. no FC-140-1083). CD4 + cells served as germline control. The exome sequencing and bioinformatics pipeline to discover somatic variants were done as described previously[13], with the same analysis pipeline as with immunogene panel data.

**RNA sequencing.** RNA was isolated using Qiagen miRNeasy micro kit (cat. no 217084) and SMART-Seq v4 Ultra Low Input RNA kit (cat. no. 634890) for Sequencing was utilized to produce cDNA libraries. cDNA fragmentation and sequencing adaptor tagging war performed using Illumina Nextera XT kit (FC-131-1096). Sequence data correction, quality trimming and read length filtering were performed with Trimmomatics using the following settings: leading: 3, trailing: 3, sliding window: 4:15 and minlen: 36. The gap-aware STAR aligner was used to perform the spliced alignment of final clean paired-end reads against the human reference genome (Ensembl GRCh38) with the guidance of the EnsEMBL reference gene models (EnsEMBL v80). The default alignment and indexing settings (2-pass per-sample) were used, except that the overhang on each side of the splice junctions was set to 99. Picard tools were used to sort the reads and to mark the reads. The reads were then assigned to genomic features using SubRead. Default parameters were used in the feature summations except that reads were allowed to be assigned to overlapping genome features. Quality control analysis was performed using the RNA-SeQC and FASTQC with default settings and using genome annotation information downloaded from EnsEMBL (v80). The obtained BAM files were normalized and the data was explored and visualized using the Qlucore Omics Explorer 3.2(Qlucore AB, Lund, Sweden).

**HLA typing method.** All the samples were typed at the Histocompatibility Testing Laboratory, Finnish Red Cross Blood Service accredited by European Federation for Immunogenetics. The HLA specificities were reported based on the current World Health Organization (WHO) nomenclature for the HLA system.

The typing for HLA-A, -B, -C and -DRB1 loci was performed using the Luminex bead array technology together with sequence-specific oligonucleotide probes (Commercial LabType kits RSSO1A, RSSO1B, RSSO1C, RSSO2B1, One Lambda, Los Angeles, CA). The bead array data were interpreted according to the manufacturer's recommendations using the HLA Fusion software 3.2 (One Lambda).

A proportion of samples was further typed by Sanger sequencing method to obtain higher resolution for the HLA type (Commercial AlleleSEQR kits 08K60-06, 08K61-06, 08K62-06, 08K63-06, GenDx, Utrecht, Netherlands). The sequencing data were analysed using the SBTengine software 3.9.0.2563 (GenDx).

**Shared epitope alleles.** Known HLA susceptibility loci, the shared epitope alleles, were defined in our study as HLA-DRB1*01, HLA-DRB1*04, HLA-DRB1*10 and HLA-DRB1*14:02 (ref. 15).

**Statistical analysis.** Bioinformatic methods are described in other sections. Statistical analyses were performed using Prism 6 for Mac OS X, Version 6.0, and R (version 3.3.1). In all analysis $P < 0.05$ were considered as statistically significant. Normality of the data was inspected graphically and tested with Shapiro–Wilk test. If one of the studied groups did not follow normal distribution, or if the group sample size was small, nonparametric statistical tests were performed. Statistical methods comprised of Mann–Whitney test, Fisher's exact test, linear regression and Spearman correlation, which were used as indicated in figure legends.

**Data availability.** The sequence data that support the findings of this study are available in ImmuneAccess with the identifier https://doi.org/10.21417/B76C7W. Other sequence data (immunogene panel, exome and RNA sequencing raw data) are only available from the corresponding author upon request, owing to regulations pertaining to the authors ethics permit and deposition of these data in public repositories.

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

## Acknowledgements

Personnel at the Hematology Research Unit Helsinki, Department of Rheumatology, University of Helsinki and Helsinki University Central Hospital, and Institute of Molecular Medicine Finland (FIMM) are acknowledged for their expert clinical and technical assistance. Sequencing library preparation, immune gene targeting, NGS, RNA sequencing, and primary data analysis were conducted by the Technology Centre of FIMM. Suvi Kyttänen, Sari Hannula, Pirkko Mattila and Matti Kankainen are acknowledged for their technical expertise in sequencing and data analysis. Cell sorting experiments were performed at Helsinki Biomedicum Flow cytometry core facility. Drs Olli Kallioniemi, Caroline Heckman and Jukka Vakkila are acknowledged for their expert comments regarding the targeted sequencing panel. This work was supported by the European Research Council (M-IMM project), Academy of Finland, Finnish special governmental subsidy for health sciences, research and training, Sigrid Juselius Foundation, Instrumentarium Science foundation, Finnish Cultural Foundation, Maire Lisko foundation, Emil Aaltonen foundation, Orion research foundation, and Finnish Cancer Institute.

## Author contributions

P.S., T.K., H.L.R., A.K., K.K., T.J., H.R., K.P., M.L.-R. and S.M. designed the study and experiments. M.L.R and R.K. recruited patients. P.S., T.K., H.R., A.K., K.K., R.K.K. and E.I.A. performed experiments. P.S., T.K., H.L.M.R., A.K., K.K. and E.I.A. analysed data. S.E. performed bioinformatic analyses. T.J. designed and supervised HLA typing and analysis. P.E., S.L., M.L., T.H., S.H., E.I.A. and J.S. designed, supervised and performed sequencing assays. P.S., T.K. and S.M. prepared the manuscript. K.P., M.L.-R. and S.M. supervised the study. All authors contributed to the writing process and accepted the manuscript.

## Additional information

**Competing interests:** S.M. and K.P. have received honoraria and research funding from Novartis, Bristol-Myers Squibb and Pfizer. S.M. has received research funding from Ariad. J.S. has received lecture fees from Roche. The remaining authors declare no competing financial interests.

