## [Peer Review File · Nature Communications]

Reviewers' comments:

Reviewer #1 (Remarks to the Author):

This article addresses an interesting and challenging concept, i.e. whether somatic mutations related to functional changes in CD8 positive T cells may contribute to the development of non-malignant inflammation, here in RA. The results are novel and interesting in many respects, but there are several issues that are unclear and some interpretations that this reviewer cannot share. Below follow some specific comments

- On p 4 and at several other places in the article, there is a confusion concerning the use of the concept "T cell clone". For example, it is stated that "the most prominent T cell clones constitute 20-55% of the CD8+ T cell population in blood. However, this statement is based on immunostaining (in flow cytometry) with anti-Vb antibodies. It is obvious (and also stated further down in the text) that many different T cell clones can be identified with the same anti-Vb antibody. If the authors want to use the word "T cell clone" for cells that share the same TCR (in this case the same sequence in the Vb), there have to be more careful with the concept of "clonality" and the nomenclature throughout the article.
- On p5 the authors describe an analysis where they relate the genotype of the MHC class II genes to the presence of clones (for the concept of clones, see comment above). However, the number of individuals from which the cells were taken is too small to permit the very definite conclusion in the text.
- In p 5 there is also a statement where "clonality" measured by flow cytometry as compared between RA patients and healthy controls. However, only 7 healthy controls were analysed and 2 of them had more than 10% CD8 cells staining with a given anti-Vb antibody. First, this material is obviously too small to permit conclusions concerning comparisons between healthy individuals and controls, second only cells from these 2 individuals were used in subsequent analysis of somatic mutations in healthy individuals respective RA patients (later in text). This small number obviously makes all conclusions on potential differences between cases and controls uncertain.
- The data presented in supplementary table 3 on clonality using deep sequencing of Vb genes are interesting and indicate a major increase in CD8 positive cells that are indeed derived from a single precursor. However, the data do not necessarily prove that the number of CD8 positive cells expressing this particular Vb sequence is the same as the size of the mRNA which is sequenced. This should be clarified.
- The major novel part of this study is presented on p 6 and following pages. Two methods were used: One targeting a selected set of immunology-related genes and one broader approach. As I can read it from the text, 10 mutants occurring exclusively in CD8 positive cells (a criterion for lineage-specific somatic mutations) in cells from RA patients and 1 mutation was seen in cells from one healthy control. The other method used cells sorted by means of anti-Vb antibodies and comprised 5 patients and one healthy control. Here, 24 somatic mutations were identified making 31 unique in total. Again the material used from controls is very limited (one patient out of 2 with Vb antibody based "clonality" used. This limits the interpretations of comparisons between cases and controls.
- The "case histories" are interesting and describe the accumulation of somatic mutations in the expanded subsets of CD8-positive cells. The observation is interesting in RA but not novel as a concept.
- On p 14 comes some of the more problematic statements (and conceptual limitations) of this study. The authors state that "Nevertheless, the genes found to be mutated in our patient material (such as PADI4 and IRF1) are most interesting as they have been shown to be closely associated with the initiation and progression of RA" These genes have indeed been identified in different GWAS investigations, but their functions in RA have not been assigned to CD8-positive cells, but rather to completely different parts of the inflammatory process in RA. This statement also reflects the fact that no efforts has been made to link the observed somatic mutations to distinct functions in normal or malignant CD8-positive T cells.

Reviewer #2 (Remarks to the Author):

Review of "Somatic Mutations in Clonally Expanded 1 Cytotoxic T Lymphocytes in Patients with Newly Diagnosed Rheumatoid Arthritis", manuscript # NCOMMS-16-13438

In the paper by Savola, Kelkka et al. lymphocyte fractions were analysed in newly diagnosed RA patients. The authors report on 46% of analysed patients with clonally expanded CD8+ T-cells, whereas no such expansion was observed in CD4+ T-cells. In further NGS analysis of selected patients, mutations in several genes in CD8+ T-cells were identified. The observed mutations were found to be stable over time in the affected patients, but these mutations were not found in CD4+ T-cells. The authors conclude that the found somatic mutation in expanded CD8+ T-cells suggest a link between autoimmunity and cancer.

The idea of the present study is very innovative. To my knowledge, it is a first study reporting on somatic mutations in clonally expanded effector T-cells in patients with non-malignant disorders. The authors collected a large cohort of well characterised patients and performed a lot of flow cytometric and sequencing analysing in order to characterise phenotype and genotype of peripheral blood lymphocytes. A very nice hypothesis has been concluded from the results obtained. However, I have serious concerns regarding the methodology applied in the study, which (from my experience) is inappropriate to achieve study goals. Therefore, the study results and their interpretations can not be accepted in the current form and should be verified by the appropriate technology.

My critique

Major points

1. The major and most important point is the definition of a T-cell clone and the method by which the clonotype repertoires were identified in the current study.

The definition of the term epitope and of the term clonotype is imprecise and must be improved. (For instance, the epitope used to describe different V-genes in the legend for supplementary figure 3). The usage of clonotype is even more confusing as it sometimes refer to just the V-genes, sometimes to V-J combination, but never to its true definition V-CDR3-J combination. Fig 4b and f display V-J combination and cannot be used to deduce if this population is monoclonal as the same V-J combination can produce a multitude of different CDR3. It has long been hypothesised and recently shown (Shanon et al, 2016) that some V-genes are positively selected for by particular HLAs. It is therefore not impossible to observe some dominance of particular V-genes in an individual even in the absence of any expansion. Also our own data show that one Vbeta gene can comprise many hundreds different CDR3 (please see the attached image demonstrating more than 350 different clones (CDR3) within V 19 gene. Even in a combination with a certain J gene (the most dominant J2-1) V19-J2-1 comprise for almost 100 different clones).

This illustrates that V-CDR3-J combination should be used to assess clonality.

Accordingly, the only method allowing the assessment of the real clonality in a certain population is V-CDR3-J sequencing. The assessment of clonality by Vbeta-staining-based flow cytometry (as performed in the current study for the most patients) is not an appropriate method and the cited Ref 15 from 1995 is unfortunately outdated.

The inappropriate method for the clonality assessment as applied in the study compromise all subsequent findings and conclusion. On the other hand, the authors report themselves on performing TCR-NGS in CD8+ T-cells of 23 patients. However, the data are shown in supp. figure only for 13 patients and only one or two clones/per patients. Why didn't the authors apply the NGS technology and the data for the whole study? Why did not they assess the T-cell clonality using NGS data?

2. According to the above mentioned sorting CD8+T-cells using Vbeta antibodies will not assure mono/oligoclonal population. Therefore, the subsequent findings on the mutations in clonally expanded CD8+T-cells are not convincing and has to be verified by appropriate technology.

My suggestion:

Findings on T-cell clonality as well as mutations in clonally expanded cytotoxic T-cells have to be verified by NGS technology. Whereas, the data on T-cell clonality have to be recalculated from CDR3-NGS data, the relationship between somatic mutations and clonal expansion can be analysed by a single cell sequencing.

Minor points:

3. Fig 1f appears to be displaying V-genes and not clones – otherwise there cannot be comparison to the flow cytometry data. It is not clear if the comparison is from one patient or pool of all observations. This must be clarified and if the data is from a pool of all patients, recalculation of frequencies must be described.

4. Term Vbeta-negative should be avoided. All TCRs contain Vbeta-genes. The authors mean for sure unexpanded populations. This should be clarified.

5. HLA-B*08 Asp9 has previously been associated with RA (Raychaudhuri et al., 2012). In the current study it appears to play no larger role indicating the complexity of the etiology. However, the lack of association to any MHC-I molecule only brings support to the notion that CD8 play no major role in the RA pathophysiology. An association is found to the previously described HLA-DRB1 does not say anything about CD8 cells, but confirms the notion that CD4 cells play a major role in RA.

6. With the exception of one patient, it is not clear what is being compared for the immunogene panel. The text indicates RA patients and healthy but does also indicate cell fractions being compared. This should be clarified.

7. The provided TCRbeta sequencing data performed on genomic DNA is filled with unproductive arrangements – in some cases these make out the most dominant clone. The reason for the unproductive arrangements can be either technical error or result of failed rearrangement during T cell development. Irrespectively of the reason, it cannot be a meaningful marker for an expanded clone. The repertoire should be analysed with these clones removed.

8. It is argued that the mutations could prevent the CD8 T cell population from contracting. CD4 T cells are also expanded in RA, but according to this study they are unaffected by mutations. If mutations are responsible for the expansion in CD8 cells, something else must drive the expansion of CD4 cells. Untreated RA is characterised by chronic inflammation. The inflamed state is the same for all patients and not just a select few and therefore a much more likely explanation for the observed expansion of CD8 T cells. I am very curious how the data look like when they are obtained after NGS analyses.

9. It would greatly improve the paper if the authors could demonstrate that hyper-expanded CD8 T cells in RA are reactive to products from the synovial fluid.

10. Lack of the overlap of the expanded T-cell clones with the published CMV clonotypes does not exclude that the expanded clones are CMV-specific. Public clonotypes are rare to our experience. Virus-specific TCR are mostly individually unique (Babel et al., 2009).

11. In the Materials and Methods it is stated: "Statistical methods comprised of Student's T test, Mann Whitney test, and Fisher's exact test, which were used as indicated in figure legends". However, no figure legend has any indication of statistical tests.

12. Given that the Vb1 staining is distinct in figure 4, it is surprising to see representation of several other V-genes after sequencing of sorted Vb1+ cells. How can this be explained? The purity is described as being nearly 100%, so the extra V-genes are not due to impurities. Is the specificity or cross reactivity of the used antibodies known?

13. A unification of the TCRV nomenclature would make reading easier.

14. It should be clarified in the Materials and Methods that the TCRVbeta antibodies used recognise the V-region of the TCRb chain, and not the CDR3 as it is currently described.

15. It should be clarified in the Materials and Methods if genomic DNA or mRNA/cDNA was used for the TCR-NGS

16. It would be interesting to see how the expanded CD8 T cells are subdivided into early (CD27+CD28+) and transitory (CD27-CD28+) memory cells as previously reported for the synovial fluid (Chao et al., 2012)

Additional comments for the authors

- 'T' is missing in TEMRA in the quadrant legend in figure 5.
- Missing x-axis label in supplementary figure 3b
- Missing '6' in supplementary figure 4
- The sentence "Oligonucleotide sequences © 2006-2011 Illumina, Inc., all rights reserved" (line 179-180) seems a copy-paste mistake.

References

- Chao et al. Characterization of Effector Memory CD8+ T Cells in the Synovial Fluid of Rheumatoid Arthritis, *Immunol* (2012) 32: 709. doi:10.1007/s10875-012-9674-3
- Raychaudhuri et al. Five amino acids in three HLA proteins explain most of the association between MHC and seropositive rheumatoid arthritis, *Nature Genetics* (2012) 44, 291–296. doi:10.1038/ng.1076
- Sharon et al. Genetic variation in MHC proteins is associated with T cell receptor expression biases, *Nature Genetics* (2016). doi:10.1038/ng.3625

Reviewer #3 (Remarks to the Author):

The manuscript by Savola et al approaches an important and interesting question related of somatic mutations driving the expansion and pathogenic function of T cells in RA subjects. Because the observations of these investigators could be of great significance in RA I do believe the level of rigor should be quite high with respect to fully expanding on this subject experimentally. Currently I believe this manuscript falls short in some respects, I have several concerns and suggestions that I will describe below.

1) The authors refer to T cells clones and clonal expansion in several setting. In my mind to be considered a clone a T cell must share the same TCR sequence. Initial studies/ figures in this manuscript are evaluating Vbeta usage and identify expansions of T cells that express the same Vbeta- in this case they are referred to as clones which I think is confusing. Particularly when sequences are later done and not all cells with common Vbeta have identical TCR. I believe the authors should be more clear with respect to the term clone.

2) Why was HLA typing done on a limited number of subjects, this is an area that would be of interest?

3) The selection of 26 clones from 23 subjects leave the reader wondering how many clones were found per subjects and if only a few subjects had many clones it would be helpful to clarify this point.

4) Clones and somatic mutations. The subjects for which somatic mutations were found are very interesting, particularly when multiple mutations are present. I think this is a point where several things are needed to enhance the importance of these findings:

a. Are the mutations all found in all cells that share a Vb or are their multiple distinct mutations within T cells subpopulations that share the same TCR? To address this I believe RNAseq with single cells or another single cell technology would be very helpful.

b. What is the specificity of these hugely expanded CD8 T cell clones with mutations- this too could be done through re-expression of the TCR and testing for binding to known viral and arthritis Ag.

c. Addressing viral specificity is important here particularly in older individuals- it seems inadequate to determine these clones are not specific for viruses based on searches of known TCR.

d. RNAseq was done on the CD8 lymphocytes (fig3) does this demonstrate the mutations in coding regions?

5) A modest concern is the use of subjects with palindromic rheumatism for the studies of the clones- 2/5 these subjects may have a different pathogenesis of disease so may not be ideal for generalization to RA and additional difference in this small group is that some are seropositive while others are not- an area where pathogenesis may differ.

Response to the reviewer comments

Revised portions in the manuscript text have been highlighted with red font for visibility.

Reviewer comments are marked in “double quotations”, and responses without quotation marks and marked with a # symbol.

Response to the Editor and all reviewers:

#We have carefully studied the comments, and present novel data to answer the criticism raised by the reviewers. With the addition of the new data, the main message of our manuscript was significantly strengthened and allows us to present a more accurate description of the data.

Firstly, we performed additional TCR beta (TCRB) chain deep sequencing (NGS) to yield data from 65 RA patients from which suitable samples were available (sorted CD8+ cells) and from 20 healthy controls. This allowed us to replace statistical analyses that were earlier based on flow cytometry data by calculations based on the more accurate sequencing data.

Next, we analysed additional RA patients (now a total of 25) and healthy controls (now a total of 20) using the immunopanel NGS sequencing assay to better understand if the described somatic mutations only occur in RA patients, or if the phenomenon can also be

seen in healthy controls. Despite good sequencing depth, novel mutations could neither be identified from the controls nor from the additional RA patients.

HLA-genotyping was extended to comprise a total of 65 RA patients and 20 healthy controls thus the updated HLA-data analysis now covers the whole obtainable cohort.

To gain better understanding of the biological processes in the expanded cell clones, we performed bulk-RNA sequencing from V β -antibody stained, flow-cytometry-sorted cell fractions in which the mutations were located. In addition, RNA sequencing was performed from polyclonal CD8⁺ T cell pool from same patients and selected healthy controls to be able to compare the properties of the expanded CD8⁺ pool to normal CD8⁺ cells.

Lastly, we have modified the whole text and paid special attention to the terminology (in particular to the usage of the word “clone” as suggested by the reviewers). In conclusion, we feel that we have been able to respond to the reviewer comments and that the novel data further strengthens our initial findings.

Reviewers' comments:

Reviewer #1 (Remarks to the Author):

“This article addresses an interesting and challenging concept, i.e. whether somatic mutations related to functional changes in CD8 positive T cells may contribute to the development of non-malignant inflammation, here in RA. The results are novel and interesting in many respects, but there are several issues that are unclear and some interpretations that this reviewer cannot share. Below follow some specific comments

- On p 4 and at several other places in the article, there is a confusion concerning the use of the concept “T cell clone”. For example, it is stated that “the most prominent T cell clones constitute 20-55% of the CD8+ T cell population in blood. However, this statement is based on immunostaining (in flow cytometry) with anti-Vb antibodies. It is obvious (and also stated further down in the text) that many different T cell clones can be identified with the same anti-Vb antibody. If the authors want to use the word “T cell clone” for cells that share the same TCR (in this case the same sequence in the Vb), there have to be more careful with the concept of “clonality” and the nomenclature throughout the article.”

#The use of “clone, clonality” is revised and used only in the context of NGS TCRB sequencing data. As the TCRB sequenced cohort was expanded to 65 patients and 20 healthy controls, most analyses are now done using TCRB sequencing data only.

- On p5 the authors describe an analysis where they relate the genotype of the MHC class II genes to the presence of clones (for the concept of clones, see comment above). However, the number of individuals from which the cells were taken is too small to permit the very definite conclusion in the text.”

#HLA genotyping as well as TCRB deep sequencing has now been performed from the total of 65 patients. Also, the control population was expanded to consist of 20 healthy controls. New analyses show that HLA-DRB1 alleles are overrepresented in RA patients when compared to the healthy controls. Since the association in the previous version of the manuscript was based on flow cytometry and thus does not represent unique T-cell clones, we repeated this analysis using the clonality index calculated from TCRB sequencing (revised Supplementary Figure 4). The flow cytometry based analysis was removed from the manuscript.

- In p 5 there is also a statement where “clonality” measured by flow cytometry as compared between RA patients and healthy controls. However, only 7 healthy controls were analysed and 2 of them had more than 10% CD8 cells staining with a given anti-Vb antibody. First, this material is obviously too small to permit conclusions concerning comparisons between healthy individuals and controls, second only cells from these 2 individuals were used in subsequent analysis of somatic mutations in healthy individuals respective RA patients (later in text). This small number obviously makes all conclusions on potential differences between cases and controls uncertain.”

#As stated above, TCRB deep sequencing has now been performed from a total of 65 patients and 20 controls. This greatly improves our possibilities to compare the different populations. With a larger healthy control population, we could no longer see the difference in clonality (defined based on NGS TCRB sequencing data) between the patients and the

controls. TCR clonality in RA patients increased with age, in line with previous observations of decreased TCR diversity with increasing age (Qi et al. PNAS 2014). Furthermore, a few of RA patients had marked CD8+ expansions (up to 51%), and similarly large expansions were not discovered in healthy controls.

With the increased number of NGS TCRB sequenced subjects we have also improved the description of clonality in the study samples. However, we agree that without the alpha chain sequencing the definition of clonality remains only “almost” conclusive. All statistical analyses have now been performed using NGS TCRB sequencing data, and we are convinced that the novel data improves the reliability of the conclusions.

“• The data presented in supplementary table 3 on clonality using deep sequencing of Vb genes are interesting and indicate a major increase in CD8 positive cells that are indeed derived from a single precursor. However, the data do not necessarily prove that the number of CD8 positive cells expressing this particular Vb sequence is the same as the size of the mRNA which is sequenced. This should be clarified.”

#All TCRB sequencing analyses (both in the initial version and in the improved version of the manuscript) have been performed using genomic DNA (now stated in the materials and methods section under the subtitle “ Sequencing assays; TCRB CDR3 deep sequencing”). In the revised version of the manuscript, the analyses have been refined to encompass only productive TCR sequences. The sequencing assay utilized genomic DNA in all cases, and it allows the quantification of all rearranged TCRB templates, and thus calculating the clone frequency is possible.

“• The major novel part of this study is presented on p 6 and following pages. Two methods were used: One targeting a selected set of immunology-related genes and one broader approach. As I can read it from the text, 10 mutants occurring exclusively in CD8 positive cells (a criterion for lineage-specific somatic mutations) in cells from RA patients and 1 mutation was seen in cells from one healthy control. The other method used cells sorted by means of anti-Vb antibodies and comprised 5 patients and one healthy control. Here, 24 somatic mutations were identified making 31 unique in total. Again the material used from controls is very limited (one patient out of 2 with Vb antibody based “clonality” used. This limits the interpretations of comparisons between cases and controls.”

#The firstly mentioned approach targeting a set of immunology-related genes was initially designed to discover mutations that are more likely to have important functions in lymphocytes and could have the potential to alter the functional properties of the affected cells. In the original manuscript we examined altogether 7 healthy controls with the immunopanel-sequencing method and identified only one somatic mutation in one of the subjects. Now we sequenced 13 additional controls, but could not identify any additional mutations in these control samples. Also the RA cohort was expanded by two patients to contain 25 immunopanel-sequenced subjects. No additional mutations could be identified in the RA patients either. Vβ-antibody-mediated cell sorting and exome sequencing was utilized in three RA patients, and these results were presented also in the previous version of the manuscript.

“• The “case histories” are interesting and describe the accumulation of somatic mutations in the expanded subsets of CD8-positive cells. The observation is interesting in RA but not novel as a concept.”

To our knowledge, there is only one known example of an autoimmune disease which has acquired somatic genetic changes: the rare autoimmune lymphoproliferative syndrome (ALPS) (Goodnow, Cell, 2007). It is characterized by lymphadenopathy and autoimmunity occurring in early childhood. The disease was first discovered to be caused by inherited mutations in the FAS gene (Fisher et al, Cell, 1995, Rieux-Laucat et al, Science, 1995), but recently, some sporadic cases of ALPS were found to harbor somatic FAS mutations in hematopoietic stem cells, which induced similar disease phenotype as seen in children with germline mutations (Dowdell et al, Blood, 2010, Holzelova et al, N Engl J Med, 2004). In other autoimmune diseases no similar somatic mutations have been found thus far. Thus, we consider that our results are of significant novelty.

“• On p 14 comes some of the more problematic statements (and conceptual limitations) of this study. The authors state that “Nevertheless, the genes found to be mutated in our patient material (such as PADI4 and IRF1) are most interesting as they have been shown to be closely associated with the initiation and progression of RA” These genes have indeed been identified in different GWAS investigations, but their functions in RA have not been assigned to CD8-positive cells, but rather to completely different parts of the inflammatory process in RA. This statement also reflects the fact that no efforts has been made to link the observed somatic mutations to distinct functions in normal or malignant CD8-positive T cells.”

#The results and functional analyses from different GWAS studies have not always been concordant and the function of these genes have not been studied in them in the CD8+ T cells.

Now we have performed RNA sequencing from the V β -antibody-stained, flow-sorted lymphocyte populations containing the mutated cells. Importantly, we were able to show that these sorted fractions, but not the highly polyclonal background population, harboured the identified mutations. Further, from the RNA sequencing data we were able to analyse which of the mutations were actually expressed, and thus have the potential to affect the cellular phenotype. Interestingly, the PADI4 mutation resulted in dramatic down regulation of the mRNA transcripts in the patient 1 with A359T mutation in his CD8+V β 13.1+ cell population. RNA sequencing data also gave us clues of how the mutations might affect the overall state of the cells (observed proliferation and survival associated signature). However, we agree that for deeper understanding of the functional role of the identified mutations, further testing is needed.

Reviewer #2 (Remarks to the Author):

Review of “Somatic Mutations in Clonally Expanded 1 Cytotoxic T Lymphocytes in Patients with Newly Diagnosed Rheumatoid Arthritis”, manuscript # NCOMMS-16-13438

“In the paper by Savola, Kelkka et al. lymphocyte fractions were analysed in newly diagnosed RA patients. The authors report on 46% of analysed patients with clonally expanded CD8+ T-cells, whereas no such expansion was observed in CD4+ T-cells. In further NGS analysis of selected patients, mutations in several genes in CD8+ T-cells were identified. The observed mutations were found to be stable over time in the affected patients, but these mutations were not found in CD4+ T-cells. The authors conclude that the found somatic mutation in expanded CD8+ T-cells suggest a link between autoimmunity and cancer.

The idea of the present study is very innovative. To my knowledge, it is a first study reporting on somatic mutations in clonally expanded effector T-cells in patients with non-malignant disorders. The authors collected a large cohort of well characterised patients and performed a lot of flow cytometric and sequencing analysing in order to characterise phenotype and genotype of peripheral blood lymphocytes. A very nice hypothesis has been concluded from the results obtained. However, I have serious concerns regarding the methodology applied in the study, which (from my experience) is inappropriate to achieve study goals. Therefore, the study results and their interpretations can not be accepted in the current form and should be verified by the appropriate technology. “

My critique

Major points

“1. The major and most important point is the definition of a T-cell clone and the method by which the clonotype repertoires were identified in the current study.

The definition of the term epitope and of the term clonotype is imprecise and must be improved. (For instance, the epitope used to describe different V-genes in the legend for supplementary figure 3). The usage of clonotype is even more confusing as it sometimes refer to just the V-genes, sometimes to V-J combination, but never to its true definition V-CDR3-J combination. Fig 4b and f display V-J combination and cannot be used to deduce if this population is monoclonal as the same V-J combination can produce a multitude of different CDR3.”

#We acknowledge our imprecise use of terminology and we have now paid special attention to correct all the terms. In addition, to better address the clonality issue, we have now performed TCRB sequencing by NGS from sorted CD8+ cells from a total of 65 RA patients and 20 healthy controls.

Figure 4 indeed showed CD8+ T-cell clones in panels 4b and 4f plotted only based on V-J genes. These figure included only V-J combinations for visual clarity, to show the highly clonal architecture of the CD8+ pool. We are aware that a true CD8+ T-cell clone is defined by VDJ-recombination and is defined by its nucleotide sequence. This fact is now taken into account in the Fig. 4 and its legend.

“It has long been hypothesized and recently shown (Shanon et al, 2016) that some V-genes are positively selected for by particular HLAs. It is therefore not impossible to observe some dominance of particular V-genes in an individual even in the absence of any expansion.”

We have now performed HLA genotyping from all RA patients from whom TCRB deep sequencing was also done (n=65). HLA-I allele frequencies in RA patients did not differ from healthy controls. Further, there was no difference in V gene family usage between patients and controls (Fig.1e). More detailed associations generally require significantly higher amounts of individuals in the study, such as in the paper by Shanon et al. Future studies will need to assess the interplay of HLA-I alleles and the CD8+ TCR repertoire in more detail.

“Also our own data show that one Vbeta gene can comprise many hundreds different CDR3 (please see the attached image demonstrating more than 350 different clones (CDR3) within V 19 gene. Even in a combination with a certain J gene (the most dominant J2-1) V19-J2-1 comprise for almost 100 different clones).

This illustrates that V-CDR3-J combination should be used to assess clonality.

Accordingly, the only method allowing the assessment of the real clonality in a certain population is V-CDR3-J sequencing. The assessment of clonality by Vbeta-staining-based flow cytometry (as performed in the current study for the most patients) is not an appropriate method and the cited Ref 15 from 1995 is unfortunately outdated.

TCRB deep sequencing data is now used to assess the clonality in all analyses. In our patient cohort however, the flow cytometry data correlates well with the sequencing data in the context of V gene usage, and seem to give good estimates of extremely big cell populations (Figure 1b). In the updated version of the manuscript, flow cytometry is only used to sort V β -enriched cell populations and to show that the mutations are restricted to a certain cell pool (certain clone/clones), and not present in all CD8+ cells.

The reference 15 (Fitzgerald et al. 1995, J Immunol) has been removed.

“The inappropriate method for the clonality assessment as applied in the study compromise all subsequent findings and conclusion. On the other hand, the authors report themselves on performing TCR-NGS in CD8+ T-cells of 23 patients. However, the data are shown in supp. figure only for 13 patients and only one or two clones/per patients. Why didn't the authors apply the NGS technology and the data for the whole study? Why did not they assess the T-cell clonality using NGS data?”

#When the prospective collection of the samples was initiated in 2011-2012, NGS TCRB sequencing was not yet feasible in large scale, not least due to its high costs. Now for the revised version of the manuscript, we extended the NGS TCRB sequencing analysis to cover all subjects from whom we had access to large enough amount of sample material. Now we have performed NGS TCRB sequencing for altogether 65 RA patients and 20 healthy controls and the novel sequence based data is used for all subsequent data analysis. In addition, it should be noted that the TCRB NGS was done from sorted CD8+ cells (not from the total T cell population) and in some cases even from the flow sorted VB expanded population giving detailed picture of CD8+ T cell clonality in our patient cohort.

“2. According to the above mentioned sorting CD8+T-cells using Vbeta antibodies will not assure mono/oligoclonal population. Therefore, the subsequent findings on the mutations in clonally expanded CD8+T-cells are not convincing and has to be verified by appropriate technology.

My suggestion:

Findings on T-cell clonality as well as mutations in clonally expanded cytotoxic T-cells have to be verified by NGS technology. Whereas, the data on T-cell clonality have to be recalculated from CDR3-NGS data, the relationship between somatic mutations and clonal expansion can be analysed by a single cell sequencing.”

#

- 1. All T-cell clonality analyses have now been calculated using NGS TCRB sequencing data.**
- 2. Unfortunately, considering the time frame for the manuscript revision and the sample availability, we do not have the practical possibility to perform single-cell RNA sequencing. Thus, the best approach for us today is to sort the cells and, thus, to enrich the cells expressing a specific V β gene and in some cases even perform TCRB NGS sequencing from the sorted clone (please see Fig 3, panel e, patient 2 as an example). The identified mutations are confirmed to exclusively exist in these sorted fractions. We show that the major clone size in TCRB-sequencing correspond**

well with the mutation variant allele frequencies in the sorted cell pool, while the polyclonal CD8+ pool that is not stained by the V β antibody does not contain mutations. Thus, we are confident that the large clone identified in NGS-B analysis and enriched by flow cytometry harbors the identified mutations.

Minor points:

“3. Fig 1f appears to be displaying V-genes and not clones – otherwise there cannot be comparison to the flow cytometry data. It is not clear if the comparison from one patient or pool of all observations. This must be clarified and if the data is from a pool of all patients, recalculation of frequencies must be described.”

#Fig. 1f indeed depicted V genes and not clones. For this revision, the figure has been redone with NGS TCRB sequencing data and terminology has been corrected. In Fig 1b, flow-cytometry populations have been compared to the sum of all cells displaying the corresponding V β gene(s).

“4. Term Vbeta-negative should be avoided. All TCRs contain Vbeta-genes. The authors mean for sure unexpanded populations. This should be clarified.”

#Yes, we meant polyclonal background cells and the terminology has been corrected and clarified.

“5. HLA-B*08 Asp9 has previously been associated with RA (Raychaudhuri et al., 2012). In the current study it appears to play no larger role indicating the complexity of the etiology. However, the lack of association to any MHC-I molecule only brings support to the notion that CD8 play no major role in the RA pathophysiology. An association is found to the previously described HLA-DRB1 does not say anything about CD8 cells, but confirms the notion that CD4 cells play a major role in RA.”

#We agree that CD4 positive lymphocytes are known to be crucial for the RA pathogenesis. However, it does not rule out that CD8+ T cells could participate as well. Originally, we performed HLA typing to see whether the patients with somatic mutations are “normal” RA patients who have typical shared epitopes. Class I alleles were analysed to better characterize the cohort. Although we have now done additional HLA typings (now total of 65 patients and 20 controls), our cohort is yet too small to achieve statistical power to detect subtle differences in HLA alleles. Overall, we think that the HLA-DRB1 dominated genetic predisposition to RA (reviewed by Kim et al. Nat Rev Rheumatol 2016) does not undermine our principal finding: the discovery of somatic mutations in expanded CD8+ cells in RA patients.

“6. With the exception of one patient, it is not clear what is being compared for the immunogene panel. The text indicates RA patients and healthy but does also indicate cell fractions being compared. This should be clarified.”

#The deep sequencing immunogene panel is a customized NGS sequencing assay that was designed to detect somatic variants in immune-related genes. From each patient and healthy control we have sorted CD8+ and CD4+ cells and sequenced those separately. For variant calling in CD8+ cells, each individual’s own CD4+ cells were used as germline controls. This allowed us to identify variants that only occur in lymphocytes that have passed thymic

selection and are mature cells. Also, this allowed us to discard all germline variants. The variant calling pipeline was also applied on CD4+ cells using CD8+ cells as germline control. This approach did not return any mutations that would have been confirmed by amplicon sequencing. All identified mutations therefore occurred in CD8+ cells. The exactly same approach was used for all healthy controls: CD8+ cells were compared with the same individual's CD4+ cells, and vice versa. We have tried to clarify this in the text (starting from page 6, line 155 and continued to page 7).

“7. The provided TCRbeta sequencing data performed on genomic DNA is filled with unproductive arrangements – in some cases these make out the most dominant clone. The reason for the unproductive arrangements can be either technical error or result of failed rearrangement during T cell development. Irrespectively of the reason, it cannot be meaningful marker for an expanded clone. The repertoire should be analysed with these clones removed.”

#Most of the unproductive rearrangements were likely to have resulted from failed somatic recombination. All of the analyses in the manuscript have now been performed using productive TCRs only.

“8. It is argued that the mutations could prevent the CD8 T cell population from contracting. CD4 T cells are also expanded in RA, but according to this study they are unaffected by mutations. If mutations are responsible for the expansion in CD8 cells, something else must drive the expansion of CD4 cells. Untreated RA is characterised by chronic inflammation. The inflamed state is the same for all patients and not just a select few and therefore a much more likely an explanation for the observed expansion of CD8 T cells. I am very curious how the data look like when they are obtained after NGS analyses.”

#We have now analysed the clonality data based on TCRB sequencing. At least in the cases of patients 1, 2, and 3, the highly expanded CD8+ T-cell clones remain relatively stable during the follow-up (revised Fig. 4). Chronic inflammation may play a part in the perseverance of these expanded CD8+ clones, but we wanted study this in more detail and performed RNA sequencing for flow-cytometry-sorted lymphocytes that contain the mutated clones. The results revealed that the expanded cell fractions express cell-division and survival-associated genes in a pattern matching with their expanded status. As presented in Fig. 3b these changes are common for all expanded cell populations.

When flow-sorted expanded cell populations were compared to the highly polyclonal background population within the same patient, we were able see differences that suggest that these populations are immunologically different from the polyclonal background. Thus, the mutations may alter the cell phenotype as the immunity related changes were unique for each patient. However, with the present data set, we cannot provide conclusive evidence to show that the mutations themselves regulate the division, survival or immunoreactivity of the mutated clones.

“9. It would greatly improve the paper if the authors could demonstrate that hyper-expanded CD8 T cells in RA are reactive to products from the synovial fluid.”

#The immune response in RA is directed against an array of different antigens, target repertoire differing between individual patients. In practice, the reactivity should be tested against each patient's own synovial antigens to ensure true self-reactivity. All our index patients responded well enough to treatment and no synovial punctures were needed for clinical reasons, and thus we could not obtain synovial fluid samples for self-reactivity

testing. Moreover, synovial fluid may not be the principal place of inflammation but the actual synovial tissue, and therefore that would be an ideal material for future studies.

“10. Lack of the overlap of the expanded T-cell clones with the published CMV clonotypes does not exclude that the expanded clones are CMV-specific. Public clonotypes are rare to our experience. Virus-specific TCR are mostly individually unique (Babel et al., 2009).”

#We agree with this comment, and the comparisons were performed mostly because possible matches would have been of interest. Now that the data is negative, the conclusion cannot be considered as conclusive. We are aware of the limitations of approach and have tried to modify this in the fifth paragraph of the discussion (end of page 14).

“11. In the Materials and Methods it is stated: “Statistical methods comprised of Student’s T test, Mann Whitney test, and Fisher’s exact test, which were used as indicated in figure legends”. However, no figure legend has any indication of statistical tests.”

#We have now checked and modified the legends to contain the information when applicable.

“12. Given that the Vb1 staining is distinct in figure 4, it is surprising to see representation of several other V-genes after sequencing of sorted Vb1+ cells. How can this be explained? The purity is described as being nearly 100%, so the extra V-genes are not due to impurities. Is the specificity or cross reactivity of the used antibodies known?”

#The impurities described are due to antibody cross-reactivity. However, 89.5% of the sorted population comprised of clones using the same V gene. The largest clone (defined by an unique TCR nucleotide sequence) in this sorted fraction made up 73% of all sequenced cells.

“13. A unification of the TCRV nomenclature would make reading easier.”

#We have now unified the nomenclature as suggested. As now most analyses have been performed using NGS TCRB sequencing data, most data is presented using the gene nomenclature and thus, we have been able to avoid the usage of V-gene families that are recognizable with the antibodies. As we still present some flow-cytometry based data, antibody-based V β names cannot be completely avoided.

“14. It should be clarified in the Materials and Methods that the TCRVbeta antibodies used recognise the V-region of the TCRb chain, and not the CDR3 as it is currently described.”

This has now been corrected in the materials and methods section.

“15. It should be clarified in the Materials and Methods if genomic DNA or mRNA/cDNA was used for the TCR-NGS”

#All immunopanel, exome, amplicon and NGS TCRB sequencing analyses were performed using genomic DNA. Only RNA-sequencing was performed on RNA \rightarrow cDNA. This has been further clarified in the materials and methods section in the subsection for “Sequencing assays: TCRB CDR3 deep sequencing” (page 20).

“16. It would be interesting to see how the expanded CD8 T cells are subdivided into early

(CD27+CD28+) and transitory (CD27–CD28+) memory cells as previously reported for the synovial fluid (Chao et al., 2012)”

#Chao *et al* describes these cells in a patient cohort that had received anti-rheumatic treatment, most commonly methotrexate. We collected our blood samples from patients that have not received any anti-rheumatic treatment and thus our data is not directly comparable. Furthermore, the key findings in Chao et al. were obtained using synovial fluid samples. Unfortunately, due to the limited sample availability we were not able to perform these analyses.

“Additional comments for the authors

- ‘T’ is missing in TEMRA in the quadrant legend in figure 5.
- Missing x-axis label in supplementary figure 3b
- Missing ‘6’ in supplementary figure 4
- The sentence “Oligonucleotide sequences © 2006-2011 Illumina, Inc., all rights reserved” (line 179-180) seems a copy-paste mistake.

#

- **T is added to the quadrant in Fig. 5.**
- **Flow cytometry data in the original Supplementary Fig.3b was completely replaced with novel figure describing NGS TCRB sequencing data and is now presented in Fig 1f.**
- **The original Supplementary Figure 4 has been replaced by Supplementary Figure 5 and now the missing number has been added.**
- **Copy-paste mistake was deleted.**

“References

Chao et al. Characterization of Effector Memory CD8+ T Cells in the Synovial Fluid of Rheumatoid Arthritis, *Immunol* (2012) 32: 709. doi:10.1007/s10875-012-9674-3

Raychaudhuri et al. Five amino acids in three HLA proteins explain most of the association between MHC and seropositive rheumatoid arthritis, *Nature Genetics* (2012) 44, 291–296. doi:10.1038/ng.1076

Sharon et al. Genetic variation in MHC proteins is associated with T cell receptor expression biases, *Nature Genetics* (2016). doi:10.1038/ng.3625”

#Thank you for pointing out these references. Especially the latter reference that actually was published after our manuscript was initially submitted to Nature Communications was highly interesting.

Reviewer #3 (Remarks to the Author):

“The manuscript by Savola et al approaches an important and interesting question related of somatic mutations driving the expansion and pathogenic function of T cells in RA subjects. Because the observations of these investigators could be of great significance in RA I do believe the level of rigor should be quite high with respect to fully expanding on this subject experimentally. Currently I believe this manuscript falls short in some respects, I have several concerns and suggestions that I will describe below.

1) The authors refer to T cells clones and clonal expansion in several setting. In my mind to be

considered a clone a T cell must share the same TCR sequence. Initial studies/ figures in this manuscript are evaluating Vbeta usage and identify expansions of T cells that express the same Vbeta- in this case they are referred to as clones which I think is confusing. Particularly when sequences are later done and not all cells with common Vbeta have identical TCR. I believe the authors should be more clear with respect to the term clone.”

#All three reviewers raised this concern and we have now extended NGS TCRB sequencing to include 65 RA patients and 20 healthy controls. Also the terminology has been straightened up.

“2) Why was HLA typing done on a limited number of subjects, this is an area that would be of interest?”

We have now also extended the HLA typed cohort to include all possible (n=65) RA patients and 20 healthy controls to better characterize the patients. As expected, DRB1 shared epitope alleles were more common in the RA patient cohort than in the healthy controls. HLA-I allele frequencies in patients and controls did not differ, as is expected based on the previous GWAS data on RA (RA genetics reviewed by Kim et al. Nat Rev Rheumatol 2016). Future studies will need to assess the interplay of HLA-I alleles and the CD8+ TCR repertoire in more detail, but detailed analyses on the subject are beyond the scope of this study.

“3) The selection of 26 clones from 23 subjects leave the reader wondering how many clones were found per subjects and if only a few subjects had many clones it would be helpful to clarify this point.”

#Thank you for pointing out this. Revised Fig. 1e provides an overview of the number of patients harbouring large clones.

“4) Clones and somatic mutations. The subjects for which somatic mutations were found are very interesting, particularly when multiple mutations are present. I think this is a point where several things are needed to enhance the importance of these findings:

a. Are the mutations all found in all cells that share a Vb or are their multiple distinct mutations within T cells subpopulations that share the same TCR? To address this I believe RNaseq with single cells or another single cell technology would be very helpful.”

We agree with the reviewer that single-cell RNA sequencing would be highly interesting to perform to solve this issue. Unfortunately, we could not perform single cell analysis for this revision due to the sample availability and time-frame aspects. However, as stated above, we have performed both amplicon and TCRB deep sequencing from expanded clones in some cases. As the variant allele frequencies from different sequencing platforms match with TCRB sequencing data and as the identified somatic mutations are restricted to the flow sorted fractions, we find it very likely that the mutations reside in expanded clones of monoclonal origin. Please see figure 3e as an example of sorted Vbeta clone from which both TCRB NGS and amplicon sequencing has been performed.

“b. What is the specificity of these hugely expanded CD8 T cell clones with mutations- this too could be done through re-expression of the TCR and testing for binding to known viral and arthritis Ag.”

#We agree with the reviewer that the knowledge of the antigen target of these clones would

be of considerable interest. We have actually been working on this for the last 2 years with our collaborators in UK (group of prof. L. Wooldridge). The array of potential targets is countless, and the question must be tackled using a hypothesis-free approach, such as a peptide-library screening. We, along with our collaborators, have tried to culture the mutated clones into monoclonal cell lines, but that has not been successful yet. Cloning the $\alpha\beta$ TCR of interest is an alternative strategy.

However, due the relatively unspecific antigen recognition properties of TCR (eg. when compared the specificity of monoclonal antibodies undergone germinal center maturation) leading to cross-reactivity (Sewell, *Nat Rev Immunol*, 2012; Wooldridge *et al*, *J Biol Chem* 2012) the results will be challenging to interpret. We will continue to work on these aspects and hope to bring more light on the issue in our future work.

“c. Addressing viral specificity is important here particularly in older individuals- it seems inadequate to determine these clones are not specific for viruses based on searches of known TCR.”

#As discussed above, we agree that this is an interesting and important aspect. We are aware that searching for public TCRs is not a comprehensive way to study this issue. However, if the approach had revealed public TCRs in the mutated clones, the data would have been of interest. Currently, the comprehensive conclusions cannot be made from this analysis, and this has been taken into account in the 5th paragraph of the discussion (starting from the end of page 14).

“d. RNAseq was done on the CD8 lymphocytes (fig3) does this demonstrate the mutations in coding regions?”

The RNA sequencing results presented in the initial submission were obtained using only CD8+ cells isolated from healthy controls. Now we used V β antibodies to enrich the mutation-containing cells and performed RNA-sequencing for these fractions. The novel data now confirm the mutations on nucleotide level and describe the expression levels of the mutated genes in the three selected index patients (Revised Fig. 3). The expression levels also allow us to assess the potential functional impact of each individual mutation on the cellular function.

“5) A modest concern is the use of subjects with palindromic rheumatism for the studies of the clones- 2/5 these subjects may have a different pathogenesis of disease so may not be ideal for generalization to RA and additional difference in this small group is that some are seropositive while others are not- an area where pathogenesis may differ.”

#We recruited all newly diagnosed RA patients who fulfilled ACR criteria and who were willing to participate. No difference was made between different RA subgroups. It is also known that in many palindromic patients the disease develops into “conventional” RA over time (Guerne & Weisman, *Am J Med*, 1992), so at diagnosis, the palindromic cases are often overrepresented.

Our cohort is in all aspects a typical RA cohort. The ratio between the seropositive and negative patients is close to what is expected, and also the shared epitope alleles are overrepresented similarly as known to be typical for RA. We agree that 2/5 is a large number, but at this point we hesitate to conclude further on this subject in the actual manuscript text.

RESPONSES TO REVIEWERS

Reviewer #1 (Remarks to the Author):

The authors have satisfactorily answered my comments. Only a minor issue remains, i.e. that the authors do not appear to be familiar with the quite extensive recent literature on autoantigens targeting by T cells in RA. This part of the Discussion would benefit from an update.

Thank you for the positive response. We have now improved the discussion on autoantigens and T cells in RA in the manuscript (p. 15, lines 369-384). Most literature on the subject presents findings relevant to CD4+ T cells, whereas CD8+ T cells have received significantly less attention.

Reviewer #2 (Remarks to the Author):

The authors performed a lot work trying to address the questions and concerns raised by the reviewers. While some points became more clear now several new issues emerged. So, unfortunately, I have now even more doubts and the data look for me as an epiphenomenon. In addition, I have serious concerns regarding the quality of the sequencing.

#As detailed below, we consider that most of the concerns that the reviewer had related to the sequencing technology were misunderstandings, and therefore, those may have caused the doubts for our results, which lead to negative decision. We have now addressed the criticism for sequencing results in detail below, and hope that they clarify our results further.

In addition, the main findings (somatic mutations in non-malignant cells), which were shown in the original manuscript, did not change during the revision process, but were further strengthened with the addition of the new data.

In general, I can summarize my concerns as following:

The paper presents the following observations: Some RA patients have expanded CD8 clonotypes, but so do some healthy donors. Some RA patients have mutations in the expanded CD8 cells which are not found in CD4 T-cells. The main message is indeed that somatic mutations are found in CD8 cells of RA patients. It is quite an intriguing observation, and through the authors try to make this observation relevant for RA, the association to RA remains speculative. To my understanding, the authors fail to demonstrate that the mutations are more than just random events.

#The reviewer has understood correctly that the main finding in our work is the identification of somatic mutations in the expanded, non-malignant CD8+ lymphocytes. We want to emphasize that identification of these mutations constitute a novel finding even though their impact on the disease remains unknown. So far somatic mutations have considered to be a hallmark of cancer,

and we widen the concept substantially by showing the presence of acquired genetic alterations also in chronic inflammation.

Furthermore, in the manuscript we do not claim that these mutations cause RA. Our main message is that the mutations can be found in RA patients and they persist during the follow-up, and therefore they are not just random events, which can be found in one blood sample. We agree, that the biological/clinical impact of the mutations needs further studies, but we want to highlight that our results pose fundamental questions for future studies in T-cell biology and in autoimmune diseases. In the abstract we have now modified the text and conclude: "In conclusion, in untreated RA patients the expanded CD8+ T-cells commonly harbor somatic mutations, and further studies are needed to understand their pathogenetic significance in RA and other autoimmune diseases." (lines 51-54)

Several observations speak for the observations being just random events: For instance, not a single mutation is shared between the 5 patients with mutations arguing against a role of any of the observed changes in RA.

#Recurrent mutation sites are not common, not even in established cancer. Even the most important driver mutations occur in genomic sites that are not shared between all patients and may occur only in a small fraction of patients (Pon & Marra, *Annu Rev Pathol* 2015, PMID: 25340638). Importantly, we did not study cancerous cells, but focused on expanded, non-malignant cells.

The lack of identification of RA associated MHC-I molecules, makes a role of CD8 cells not very likely. It is true that the cohort might be too small to identify such an association, but that does make an association likely.

Extensive work on RA genetics (good review by Kim et al. *Nat Rev Rheumatol*, 2017) has shown the CD4-MHC-II axis as the major genetically determined inherited component in RA, and we have no intention to challenge this fact. However, the role of CD8+ T cells is also recognized in RA (see the recent review Petrelli, *Nat Rev Rheumatol* 2016) and should not be underestimated. We have now discussed this reference in our revised manuscript (p.16, row 397-400).

Fig 1d: That evenness increase with age in RA patients is interesting, as this indicate that there is no expansion of particular clones.

#In Fig. 1d. age is plotted against the clonality index, not evenness. The figure shows that clonality increases with age, i.e. older patients' CD8+ cells are more oligoclonal and have more large clones than younger patients' cells. The clonality index used in our analysis should not be mixed with the concept of evenness (discussed in more detail below).

It also speaks against the role of CD8 cells in RA.

CD4 cells are known to be instrumental for the development of RA. As mentioned above, the current knowledge of CD8+ lymphocytes in RA

development and regulation has been nicely reviewed by Petrelli & van Wijk (Nat Rev Rheumatol, 2016). Further, as previously reported by our group, LGL leukaemia patients with CD8+ disease clone often display RA as a comorbidity (Koskela et al. N Engl J Med, 2012), whereas RA is rarely observed in patients with NK cell expansion, which further supports the role of CD8+ lymphocytes in RA. Thus, CD8+ cells in RA should not be totally ignored.

Although it could be a nice explanation, I did not see any association between clonality and mutations. Table 1 represents 25 RA patients with repertoire clonality ranging from 0.09 to 0.63 and median clonality is actually not different from HC. From 4 patients showing mutations, 2 patients did not show high level of clonality (Index 0.2). In contrast there are many other RA patients with a much higher clonality index and no mutations.

#We show that the clonality indexes of healthy controls' CD8+ cells do not have a statistically significant difference when compared to RA patients. The number of patients and healthy controls with mutations was relatively low in our study (5 patients and 1 healthy control with mutations). Thus, the possible association of mutations with clonality remains inconclusive and should be addressed in future studies. We hypothesize that chronic inflammation, which may vary between patients, may be a key difference explaining this (p.14-15, rows 347-367)

In addition, It is not clear for me, for what reasons TCR-NGS was performed in 62 patients. Only 25 patients were characterized in details. And in only in 5 mutations analyses were performed.

#Additional TCR-NGS was performed because it was requested by the reviewers. Our initial analyses utilized FACS data, and the reviewers concluded that FACS is not sufficient to define a T-cell clone. Also, more extensive TCRB sequencing allowed us to analyze if clonality was associated with clinical parameters, such as age.

The following quote is from the first review round from the reviewer no. 2:

“The inappropriate method for the clonality assessment as applied in the study compromise all subsequent findings and conclusion. On the other hand, the authors report themselves on performing TCR-NGS in CD8+ T-cells of 23 patients. However, the data are shown in supp. figure only for 13 patients and only one or two clones/per patients. Why didn't the authors apply the NGS technology and the data for the whole study? Why did not they assess the T-cell clonality using NGS data?”

Further, the reviewer has misunderstood that the mutation analysis was only performed in 5 patients. We analyzed samples from 25 patients and 20 controls with the deep sequencing immunogene panel (both CD8+ and CD4+ fractions) covering 1000 genes and NOT only 5 patients as the reviewer states. All the mutations were also confirmed with the other method (Amplicon sequencing) and also analyzed from index patients in the follow-up samples.

I have serious concerns with regard to the found abundance of some CD8+ clones. Figure 3 demonstrates two highly abundant clones making up to 80 % of

all CD8 T-cells (each clone comprises about 40% of all CD8 T-cells). This is a very strange observation and suggest a monoclonal process like we observe in lymphoma. Were the authors not surprised? Also clonal expansion of up to 26% in HC is unusual. Considering well-known technological challenges of the TCR-NGS technology, I am wondering, what QC measures were performed to rule out possible sample/reagent impurity and what is a cross sample overlap.

#These results were surprising. In Fig. 3 we show that in patient no. 1, the two major clones make up approximately 60% of all CD8+, not 80% as stated by the reviewer. However, large CD8+ expansions are not uncommon even in healthy subjects (Fitzgerald et al, J Immunol 1995; Posnett et al, J Exp Med 1994; Khan et al, J Immunol 2002; Degauque et al, PlosOne, 2011). Further, clinical examinations confirmed that this patient does not have lymphoma.

Regarding the quality of the sequencing, we are confident that the quality is good. TCRB NGS analysis has been done with the Adaptive Biotechnologies ImmunoSeq platform and it involves detailed QC measurements. The sequencing technology of Adaptive Biotechnologies has been applied in a myriad of high-quality studies. (<http://www.adaptivebiotech.com/publications?view=research>).

In addition, we have analyzed both the diagnostic phase as well as follow-up samples from these patients (shown in figure 3), and the same T cell clone can be observed in both samples analyzed at different time-points. Therefore, it is absolutely no question of any sample/reagent impurity or cross sample overlap. Furthermore, the mutation VAFs correspond well the clone size determined by the TCRB deep sequencing.

Nevertheless, the question still remains: Are CD8 relevant in RA? (though not the focus of the paper, it would render credibility to the relevance of the observed mutations). Are the observed mutations relevant for RA? Alternatively, the paper might carry some weight if it can be made probable that it is not just a set of random observations.

The impact of CD8+ cells in RA is discussed already above. We also want to highlight that the main finding i.e. identification of somatic alterations in CD8+ cells from several individuals is important even without direct association to the disease phenotype.

Further points:

It is clear that the patients with the highest number of mutations also are the oldest. This offers an alternative explanation of age and random events as the cause of the reported observations.

Age	# mutations Immunopanel	# mutations Exome
75	4	12
72	3	0
44	0	0
74	1	9

#It could be that the patients with the highest number of mutations are older, but it does not change our principal findings. Somatic mutations in mature T-cells have not been described before, and the possibility that they may accumulate in the aging person is interesting, and poses many questions for future studies. Similarly clonal hematopoiesis (somatic mutations occurring mostly in myeloid cells derived from aging stem cells), which is a well-accepted phenomenon increases with age and that has been associated with many clinical phenotypes (Jaiswal et al NEJM 2014, Genovese et al NEJM 2014). Moreover, we would like to point that patient 3, who was 44 at diagnosis, had 3 somatic mutations in his CD8+ cells, and not 0 as stated in the table above.

To my understanding, the used clonality index is in reality Pielous' evenness (see 10.1371/journal.pone.0049024 and 10.1016/0022-5193(66)90013-0). Though evenness and diversity are related entities, they are not the same and it is probably more informative to use a diversity index such as Shannon or Berger-Parker. It is none the less cause of great confusion to use the word 'clonality' which indicates some richness based diversity, over the correct evenness or equality. The used clonality index does not indicate that older individuals harbour more and larger clones as stated on p 15, line 350-351.

It is an unusual observation that the evenness of the TCR repertoire increases with age, as it is known that that the age dependent contraction of the T-cell pool allows few clonotypes to dominate, which in turn results in a decreased evenness. Perhaps the donors were overall not that old that this effect is apparent.

#The clonality index used in our study is calculated by Adaptive Biotechnologies, and it is essentially based on Pielous' evenness and Shannon's entropy: clonality = 1 - Pielous' evenness. According to Adaptive, these are calculated with the following formulas:

$$\text{Entropy} = H = - \sum_{i=1}^N p_i \log_2(p_i)$$

$$\text{Clonality} = 1 - \frac{H}{\log_2(N)}$$

p_i is the proportional abundance of Rearrangement i , and N is the total number of Rearrangements

This clonality index takes into account different sequencing depths and sample amounts (the total amount of rearrangements), and thus it is a more reliable parameter to use than Shannon's entropy. Shannon's entropy is heavily affected

by the amount of sample sequenced (=the number of templates), as shown by the plots below:

Large clonality indexes, such as 0.6, indicate that the sample is more clonal than a sample with a clonality index of 0.2.

In our manuscript, this is presented on p. 21-22, rows 517-520:

“Clonality was calculated according to the formula:

$$Clonality = 1 - \frac{-\sum_{i=1}^N p_i \log_2(p_i)}{\log_2(N)}$$

where p_i is the proportional abundance of the rearrangement i , and N is the total number of rearrangements. The numerator of the equation is Shannon’s entropy.”

The reported association between clonality (evenness) and clinical parameters on p. 5 are quite poor. That the association is significant does not necessarily make it relevant.

#We agree that the associations are relatively poor, and we do not claim that these findings are relevant in the text.

That no CMV specific TCR could be found in a data base does not demonstrate that the TCRs are not CMV specific. The entire section (p. 12, line 291-299) serves no purpose and should be removed.

#We agree that a querying against public CMV- and other virus-specific clonotypes is not sufficient to rule out that these cells could target CMV. However, it does not prove it to other direction either. The section can be removed if considered important.

As we reported in the previous review, it is indeed an intriguing observation that somatic mutations are found in the CD8 cells but not the CD4 cells. It can be assumed that the mutation rate per cell division is equal for CD8 T-cells and for CD4 T-cells. This begs the question why the mutations accumulate in CD8 cells only. Especially given the fact that RA is known to involve activation, and therefore also proliferation, of CD4 T-cells. The reason for the accumulation of mutations in one cell population but not in the other is outside the scope of the paper, but the observation forms the basis of speculations of the possible association to RA. It is still unclear to what extent CD8 T-cells play a role in RA - are the cells merely passing through the synovium or have they actively taken hold there. If it is the latter, one would expect to see a large overlap of CD8 T cells from different joints, just as has been repeatedly shown for CD4 T-cells.

The main function of CD8+ cells is to kill cells that are infected or that are developing malignant properties. During infection, they need to react rapidly and expand to localize and kill the target cells via a cell-to-cell contact. Thus, these cells undergo fast and potent expansion upon stimulation. The CD4+ cells' main function is to support different arms of adaptive immunity and they can multiply their impact on the target mechanism efficiently via secreting cytokines. They do not need to proliferate as much as CD8+ cells, but they can enhance the response by other means. When taking this into account, we find it quite logical that CD8+ lymphocytes form clonal expansions and carry mutations that may have arisen during the clonal expansion.

As already stated above, the importance of CD8+ T cells in autoimmune arthritis is established (Petrelli & van Wijk, Nat Rev Rheumatol, 2016). Therefore, it would be extremely interesting to analyze CD8+ TCR repertoire in different RA affected joints. Unfortunately (but luckily for the patients), it is extremely difficult to obtain sample material from such cases due to the excellent treatment options available today.

The caption to Figure 6: "In our cohort CD4+ cells presented more equally distributed T cell repertoire than CD8+ T cells". I fail to find these data anywhere, and am left wondering how this observation compares to other observations of RA patients.

#This caption was based on the fact that according to FACS data, CD8+ T-cells had a more skewed TCR variable beta repertoire than CD4+ T-cells. Even though FACS does not distinguish unique T-cell clones, it does show if the variable beta region usage is skewed in different cell populations. Skewed variable beta region usage indicates that the sample is more clonal than a sample with a balanced repertoire. This data is shown in Supplementary fig. 1, and the phrasing in the caption of Figure 6 has been changed.

Minor points:

It is not clear if and how the TCR NGS data is normalised for clonality (evenness) analysis.

#This is shown on p. 21-22, row 518, in the Methods –section of the manuscript, and explained in more detail above.

Table 1 caption points to supp table 1, but the true table is supp table 2.

Yes, this is correct. We apologize for the mistake, and it has been corrected.

Largest clone size must mean two different things in Table 1 and in Supp Table 2, as the values do not at all match.

In Table 1 we present data from NGS TCRB sequencing, while Suppl Table 2 contains data that was collected from initial flow cytometry screen. For clarity we now modified the Suppl Table 2 to contain NGS TCR sequencing data.

Reviewer #3 (Remarks to the Author):

The authors have markedly improved this manuscript with the additional subjects studied and the sequencing approaches that have been applied. I think it would be helpful to discuss the significance of the differences seen between health and disease, for example the number of subjects with clonal expansions does not differ significantly between health and Ra (based on Chi-squared analysis). This is fine if it is stated clearly, since the finding of interest in fact is the number of mutations found among CD8 T cells that undergo clonal expansion.

#We thank you for the positive comments. The text is now further clarified as suggested (p.14-15, rows 347-367).

I do think the manuscript would be much improved with a sense for the specificity of these CD8 clones (even by demonstrating a lack of specificity to common viral Ag (CMV or EBV for example)).

We agree that the antigen target of the mutated cells would be interesting to know. As stated in our earlier responses, this has been attempted, but even with expert collaboration this could not be done, because the mutated clones failed to proliferate enough in culture to perform the analysis. Also, we do not have access to a large enough number of primary cells to perform the analysis, because studying possible auto- and viral antigens requires testing for hundreds or thousands of different peptides. This is discussed in the manuscript (p.15, rows 369-384)

A minor concern is the difference between the ratio of male :female subjects in the two groups- it would be helpful to do an analysis related whether this makes a difference.

We agree with the reviewer that in an optimal case healthy controls would present identical sex ratio with the patient cohort. Our main finding is the

mutations and those were identified in the expanded cell populations. As age is known to be associated with increased clonality, we firstly selected the controls to match with the age distribution of the RA cohort.

Reviewers' comments:

Reviewer #4 (Remarks to the Author):

The manuscript of Savola et al describes an analysis of T cells from RA patients and the identification of somatic mutations in expanded cell populations.

I am asked to review the sequencing data:

The authors sequence purified DNA to high depth using standard methods and validated all mutations using amplicon sequencing. Furthermore, selected samples were resequenced by exome sequencing and verified mutations found by targetseq. In addition, many mutations were also identified in transcriptome sequence. The mutations are all in different genes and not likely to be the consequence of mis-alignment.

For mutation calling methods the authors merely cite this paper:

Koskela, H. L. et al. Somatic STAT3 mutations in large granular lymphocytic leukemia. *N Engl J Med* 366, 1905-1913, (2012).

Have the methods not changed since 2012? At least a sentence or two describing the programs used (GATK, Mutect etc.) would be informative. Preferably the full methods should be given.

IGV screen shots of at least selected mutations would be useful to see the coverage and sequence context.

Silent and non-coding mutations that are not present in the CD4 cells would still be considered somatic. Although excluded by the authors pipeline, they would be informative and should also be present in increased number in the RA patient DNA. Furthermore, these mutations may have functional consequences and their presentation would add to the paper.

Are any of the mutations reported in COSMIC as previously seen in cancer cells?

The authors could cite the association of somatic mosaicism in blood cells with aging as an example of somatic events not necessarily associated with cancer. For example, (*Nat Genet.* 2012 May 6;44(6):651-8. doi: 10.1038/ng.2270. PMID: 22561519). Is there any evidence of CNVs in the RA cells? This could be examined in the exome sequences.

Minor points

Supplemental figure 8 Misspelling in title Flow-cytometriy

Supplement p 27 Immunogene panel sequnecing misspelling of sequencing

Reviewer #4 (Remarks to the Author):

“The manuscript of Savola et al describes an analysis of T cells from RA patients and the identification of somatic mutations in expanded cell populations.

I am asked to review the sequencing data:

The authors sequence purified DNA to high depth using standard methods and validated all mutations using amplicon sequencing. Furthermore, selected samples were resequenced by exome sequencing and verified mutations found by targetseq. In addition, many mutations were also identified in transcriptome sequence. The mutations are all in different genes and not likely to be the consequence of mis-alignment.

For mutation calling methods the authors merely cite this paper:

Koskela, H. L. et al. Somatic STAT3 mutations in large granular lymphocytic leukemia. *N Engl J Med* 366, 1905-1913, (2012).

Have the methods not changed since 2012? At least a sentence or two describing the programs used (GATK, Mutect etc.) would be informative. Preferably the full methods should be given.”

#The analysis pipeline has been updated after 2012, and the Methods –section now describes it in more detail (starting from page 25, rows 598-614) as suggested correctly by the reviewer

“IGV screen shots of at least selected mutations would be useful to see the coverage and sequence context.”

#Thank you for the insight. We now provide IGV screenshots from selected mutations including 3 mutations (*PADI4*, *SLAMF6* and *IRF1*) that were discovered using the immunopanel approach and 3 mutations (*PTPRO*, *CDK12* and *PLRG1*) that were found in exome sequencing data in the supplementary material (Supplementary Fig. 8)

“Silent and non-coding mutations that are not present in the CD4 cells would still be considered somatic. Although excluded by the authors pipeline, they would be informative and should also be present in increased number in the RA patient DNA. Furthermore, these mutations may have functional consequences and their presentation would add to the paper.”

We agree that the full landscape of somatic mutations including silent and non-coding mutations would be of interest. However, we chose a targeted approach (the immunogene panel) to achieve higher sequencing depth than genome-wide or exome-wide sequencing, and thus the whole genomic landscape of somatic mutations cannot be assessed. Also, this approach did not assess non-coding regions except for UTRs. Our study focused on non-synonymous mutations because they are more likely to have direct biological consequences than silent mutations. Further, all discovered non-synonymous mutations were validated with Amplicon sequencing to avoid technical artefacts. To study silent and non-coding mutations in this study, we believe that a similar, consistent approach would be appropriate, but it is beyond the scope of this study. However, the subject is intriguing, and we hope to investigate it in the future.

“Are any of the mutations reported in COSMIC as previously seen in cancer cells?”

#We have now queried all of the mutations that were validated by amplicon sequencing against the COSMIC database. Four of the mutations that result in the same amino-acid change have been observed in cancer samples.

Gene	Mutation	Genomic coordinates (hg19)	No. of samples with the same mutation in COSMIC	Tissue	Coordinates in COSMIC	COSMIC ID
CRYBB2	R160C	22:g.25627599C>T	1	Central nervous system	22:g.25627599C>T	COSM3405558
IRF1	G231E	5:g.131821384C>T	1	NS	5:g.131821384G>A	COSM5867257
CLEC10A	A235T	17:g.6978758C>T	1	Endometrium	7:g.6978758G>A	COSM983742
SMARCD1	L851W	4:g.95201870T>G	1	breast	4:g.95201870T>G*	COSM213551

*The protein amino-acid change in COSMIC was L849W, but for the canonical transcript it is L851W.

In addition, COSMIC reported three other mutations affecting the same amino-acid, but the effect on the protein was different.

Gene	Mutation	Genomic coordinates (hg19)	No. of samples with the same mutation in COSMIC	Mutation reported in COSMIC	Tissue	Coordinates in COSMIC	COSMIC ID
PTPRO	M665L	12:g.15677849A>T	1	M665I	endometrium	12:g.15677851G>A	COSM937606
CDYL	A190G	6:g.4892329C>G	1	A190V	large intestine	6:g.4892329C>T	COSM3353854
SLAMF6	F238C	1:g.160460409A>C	1	F238F	Stomach	1:g.160460408C>T	COSM4024184

These tables are now presented as Supplementary table 7, and mentioned in the text (p. 8 rows 203-206). According to COSMIC, none of these mutations were recurrent hotspot mutations in cancer.

“The authors could cite the association of somatic mosaicism in blood cells with aging as an example of somatic events not necessarily associated with cancer. For example, (Nat Genet. 2012 May 6;44(6):651-8. doi: 10.1038/ng.2270. PMID: 22561519).”

#Thank you for highlighting this study, which describes that somatic clonal mosaicism increases with age and is associated with solid tumors, and not only hematological malignancies. We added this citation to the manuscript (p.13, rows 324-325)

“Is there any evidence of CNVs in the RA cells? This could be examined in the exome sequences.”

#Exome sequencing data is available in three cases. The sequenced cells were expanded CD8+ cells (sorted with V α antibodies via flow cytometry), and CD4+ cells. When the CD4+ and the expanded CD8+V α populations from the same patient were compared, no CNVs were detected except for deletions in the chromosome 7 in the TCR gene locus. These results are likely due to the physiologic, somatic recombination of the TCR. The high clonality of the sorted CD8+ cells makes the detection of this variation possible. No other CNVs were detected. The CNV calling pipeline has been described previously in more detail by Eldfors *et al* (Leukemia 2017, PMID: 27461063).

“Minor points

Supplemental figure 8 Misspelling in title Flow-cytometriy

Supplement p 27 Immunogene panel sequnecing misspelling of sequencing”

#These mistakes have now been corrected in the Supplementary Appendix.

REVIEWERS' COMMENTS:

Reviewer #4 (Remarks to the Author):

Thank you for the careful consideration of each of my suggestions. They were all addressed in a satisfactory manner.

REVIEWERS' COMMENTS:

Reviewer #4 (Remarks to the Author):

Thank you for the careful consideration of each of my suggestions. They were all addressed in a satisfactory manner.

#Thank you for your insights for our paper.